# Highly Compressed Tokenizer Can Generate Without Training

Lukas Lao Beyer [1]   Tianhong Li [2]   Xinlei Chen [3]   Sertac Karaman [1]   Kaiming He [2]

## Abstract

Commonly used image tokenizers produce a 2D grid of spatially arranged tokens. In contrast, so-called *1D* image tokenizers represent images as highly compressed one-dimensional sequences of as few as 32 discrete tokens. We find that the high degree of compression achieved by a 1D tokenizer with vector quantization enables image editing and generative capabilities through heuristic manipulation of tokens, demonstrating that even very crude manipulations — such as copying and replacing tokens between latent representations of images — enable fine-grained image editing by transferring appearance and semantic attributes. Motivated by the expressivity of the 1D tokenizer's latent space, we construct an image generation pipeline leveraging gradient-based test-time optimization of tokens with plug-and-play loss functions such as reconstruction or CLIP similarity. Our approach is demonstrated for inpainting and text-guided image editing use cases, and can generate diverse and realistic samples without requiring training of any generative model. Code is available at https://github.com/lukaslaobeyer/token-opt.

## 1. Introduction

A standard modern image generation pipeline consists of a tokenizer — an autoencoder which embeds an image into a compact latent space — as well as a generative model operating on this latent space. Commonly, this generative model leverages some form of iterative refinement such as diffusion (Rombach et al., 2022; Peebles & Xie, 2023), autoregression (Esser et al., 2021; Ramesh et al., 2021; Razavi et al., 2019), masked modeling (Chang et al., 2022), or a combination of those approaches (Li et al., 2024b; Zhou et al., 2025; Xie et al., 2024).

[1]MIT LIDS [2]MIT CSAIL [3]Meta FAIR. Correspondence to: Lukas Lao Beyer <llb@mit.edu>.

*Proceedings of the 42nd International Conference on Machine Learning*, Vancouver, Canada. PMLR 267, 2025. Copyright 2025 by the author(s).

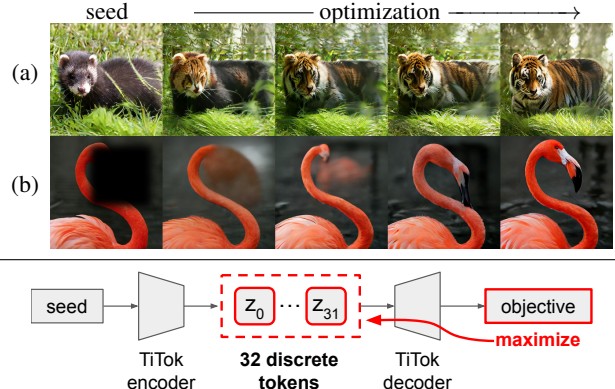

*Figure 1.* **Without further training, a pretrained tokenizer can perform generative tasks such as** (a) **text-guided editing and** (b) **inpainting**. Because the highly compressed latent space of 1D tokenizers enables generative capabilities via direct token manipulation, we generate using test-time optimization of tokens with (a) CLIP similarity or (b) reconstruction objectives, without training any dedicated generative model.

While the generative models used in such an image generation pipeline have received much attention, recently introduced *1D tokenizers* highlight that highly compressed latent spaces provide significant leverage in improving generation performance. For example, the TiTok 1D tokenizer can achieve an order of magnitude runtime reduction without compromising sample quality by representing an image in as few as 32 discrete tokens (Yu et al., 2024b).

We argue that scaling image tokenizers to higher compression ratios requires powerful decoders that may themselves be viewed as generative models. As a thought experiment, consider a tokenizer that encodes an image into just a *single* discrete token from a finite codebook. Then, the decoder for such a "tokenizer" is faced with a generative modeling task, if its output is to be successful in capturing the data distribution. Supposing we had a decoder for this single-token tokenizer at hand, using it as a generative model would be straightforward: an image could be generated by sampling a random token from the codebook and decoding it. Given the very high compression offered by 1D tokenizers, we wish to find out to what extent these tokenizers can indeed be viewed as generative models.

Through a series of experiments, we demonstrate that simple latent space manipulations of tokens can result in im-

age editing capabilities typically associated with generative models. Building upon this insight, we find that the latent space of 1D tokens is amenable to straightforward gradient-based optimization of a variety of objectives. By optimizing a reconstruction loss, a pretrained tokenizer can perform inpainting tasks; similarly, optimizing CLIP similarity (Radford et al., 2021), the tokenizer can perform text-guided image editing (Figure 1).

Combining the text-guided token optimization with retrieval from a small set of seed images, we find that it is possible to generate realistic and diverse samples without training any dedicated generative model.

## 2. Background

**Visual tokenization.** Many image generation methods rely on *tokenization* to compress input images into a lower-dimensional latent space for efficiency. A variational autoencoder (VAE) (Kingma & Welling, 2014) can be used to produce continuous tokens to be modeled with diffusion (Rombach et al., 2022). To further regularize the tokenizer's latent space and ease the task of modeling distributions over tokens autoregressively (Sun et al., 2024; Li et al., 2024b), a VQ-VAE (van den Oord et al., 2017; Razavi et al., 2019; Mentzer et al., 2024; Yu et al., 2024a) applies vector quantization to produce discrete-valued tokens. Perceptual (Zhang et al., 2018) and adversarial losses during training (Esser et al., 2021; Chang et al., 2022) allow for visually compelling reconstructions under strong lossy compression. Note that most image tokenizers produce a *2D grid of latents* due to their convolutional (Chang et al., 2022) or vision transformer (Yu et al., 2023; Cao et al., 2023) architectures.

**1D tokenization.** The spatial structure imposed by standard tokenizers producing a 2D grid of latents can prevent each individual token from describing global attributes of the input image, instead containing information only about a limited spatial extent (Cao et al., 2023; Rao et al., 2021). In contrast, the TiTok (Yu et al., 2024b) *1D tokenizer* learns a sequence of latents with no particular spatial arrangement. This allows each token to capture global information about the input image and enables extremely high compression ratios (e.g. 32 discrete tokens) because global redundancies in the input can be exploited. Further background is provided in Section A.

**Image generation without training.** Similar to our proposed text-guided image editing approach, VQGAN-CLIP (Crowson et al., 2022) performs test-time optimization of tokens, but is limited by its use of 2D latents and need for complex augmentation pipeline. It is applied in style transfer or artistic image generation settings where these limitations are less apparent. To enable text conditioning, other methods attempt CLIP-guided search (Galatolo. et al.,

2021) or gradient-based optimization (Patashnik et al., 2021) in the latent space of a pretrained GAN. Another approach is to represent visual information in the embedding space of frozen text-to-image models (Gal et al., 2023; Wang et al., 2025), enabling image generation with rich visual information as part of the input prompt. However, we focus on the setting where only a tokenizer, not a full-blown generative model like a GAN or text-to-image model, is available.

**Unconditional generation.** Unconditional generative modeling is attractive due to the difficulty associated with large-scale annotation. Many generative models benefit greatly from class labels available at training time (Dhariwal & Nichol, 2021; Bao et al., 2022), although recently the gap in performance has been bridged with the help of self-supervised pretrained encoders (Li et al., 2024a). Alternatively, *semi-parametric* generation assumes access to a dataset of images at test time (Casanova et al., 2021; Bordes et al., 2022; Blattmann et al., 2022; Sheynin et al., 2023; Zhou et al., 2023). In our case, we view the pretrained tokenizer as an unconditional generative model in the semi-parametric setting, which we can guide with the desired conditioning signal at test-time.

## 3. Latent Space of 1D Tokens

In this section we perform a series of simple experiments using a pretrained TiTok (Yu et al., 2024b) 1D tokenizer to better understand the latent space of 1D tokens and determine whether it provides a useful representation for latent-space editing of images.

### 3.1. Token Position Is Key to Token Semantics

Since the TiTok latents receive a positional encoding, one might wonder whether the 1D tokens disentangle semantics in the sense that certain token positions correspond to certain interpretable, high-level attributes. To investigate this question, we propose to analyze the TiTok encodings of several different partitions of the ImageNet validation dataset. Since there is no commonly agreed upon disentanglement metric (Locatello et al., 2019), in the following we introduce a per-token-position importance metric that we later verify to closely align with experimental results.

Start by tokenizing each image $\mathcal{I}$ in the dataset with the 1D tokenizer, obtaining the tokenized representation $\mathbf{x} := \text{Tok}(\mathcal{I}) = [x^{(1)}, ..., x^{(K)}]$, where $K$ denotes the number of image tokens produced by the tokenizer. Each token $x^{(k)} \in \{1, ..., |\mathcal{D}|\}$ refers to a particular $D$-dimensional entry $\mathbf{z}^{(k)} := \mathcal{D}[x^{(k)}] \in \mathbb{R}^D$ of the codebook $\mathcal{D}$.

Next, create a set of partitions $\{\mathcal{P}_1, \mathcal{P}_2, ...\}$ of the dataset. Each partition $\mathcal{P}_i$ splits the dataset into $N_i$ high-level classes $\mathcal{C}_{i,1}, ..., \mathcal{C}_{i,N_i}$ described by text prompts. The class assignment is computed automatically via CLIP similarity.

| Classification Prompts | Token Importance | Token Pos. (Attribute) | Token Perturbation Result |
|---|---|---|---|

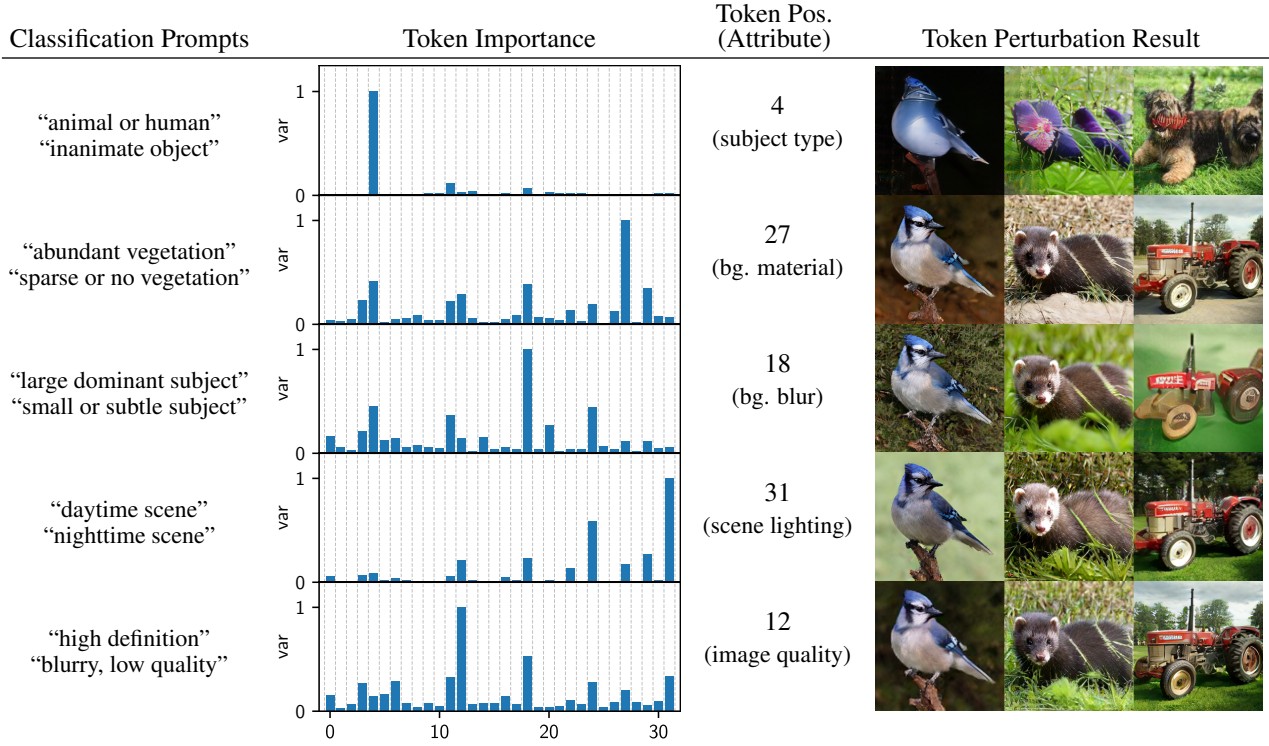

*Figure 2.* **Certain high-level attributes are represented by particular token positions.** We verify this using two independent experiments. First, we partition the ImageNet validation set into different sets of classes automatically labeled according to CLIP prompts (first column), observing for each partition that distinct token indices show the greatest variance when comparing averaged features across classes (second column; normalized). Second, we perturb token positions one-by-one (selected token positions shown in third column), visualizing the perturbation that maximizes L1 difference with respect to the unperturbed image (fourth column). **The token positions identified through the token importance experiment are consistent with the attributes altered by the token perturbation experiment.**

In order to determine which tokens contain the most information with respect to a particular partition, we will compute class-specific statistics for each token $\mathbf{z}^{(1)}, ..., \mathbf{z}^{(K)}$. Concretely, we fit a Gaussian to each set of features, yielding a distribution $\mathcal{N}(\boldsymbol{\mu}_{i,j}^{(k)}, \boldsymbol{\Sigma}_{i,j}^{(k)})$ describing feature statistics of a particular class $\mathcal{C}_{i,j}$ for each of the $K$ tokens. Given some measure of similarity between these statistics we can now determine which tokens vary the most under a particular partition. We choose to use the sample variance of the mean $\boldsymbol{\mu}_{i,j}^{(k)}$ computed across the classes of each partition,

$$g_{\mathcal{P}_i}^{(k)} = \left\| \mathrm{cov}\left( \left\{ \boldsymbol{\mu}_{i,j}^{(k)} \mid j \in \{1, ..., N_i\} \right\} \right) \right\|_2. \quad (1)$$

The leftmost two columns in Figure 2 show the result of this per-token inter-class variance computation for the pretrained TiTok-L-32 tokenizer ($K = 32$, $|\mathcal{D}| = 2^{12}$). Surprisingly, we find that for a number of different class partitions there exists significant disentanglement across token position in the sense that, for a given partition, a certain small number of token positions display a large difference in per-class mean statistics while most others do not.

### 3.2. Token Perturbations Can Achieve Meaningful Edits

The observation that a limited number of token positions vary for a given class partition suggests a crude way of editing images: is it possible to modify a certain high level attribute by directly manipulating the token at the most relevant position? We will attempt to answer this question by considering single-token edits of the following form.

$$\underset{x^{(k)} \in \{1,...,|\mathcal{D}|\}}{\arg\max} \ \ell\left( \mathrm{Dec}(\mathrm{Replace}_k(\mathrm{Tok}(\mathcal{I}), x^{(k)})), \mathcal{I} \right) \quad (2)$$

Here, Dec decodes a sequence of 1D tokens back into an image, and $\mathrm{Replace}_k([x^{(1)}, ..., x^{(K)}], y) := [x^{(1)}, ..., x^{(k-1)}, y, x^{(k+1)}, ..., x^{(K)}]$ returns its first argument with the $k$th input token $x^{(k)}$ replaced by its second argument $y$. By maximizing the loss criterion $\ell(\cdot, \cdot)$ we can therefore find the replacement token at position $k$ which causes the greatest change in output according to $\ell$.

Thanks to the modest codebook size of $|\mathcal{D}| = 4096$ we can easily solve the above optimization via exhaustive search. An L1 loss criterion is used so that the discovered single-token edits maximize visual difference.

| Attribute | background blur | scene lighting | sharpening | colorization | pose |
|---|---|---|---|---|---|
| Tokens | 18 | 31 | 12 | 12, 24, 27 | 21 |

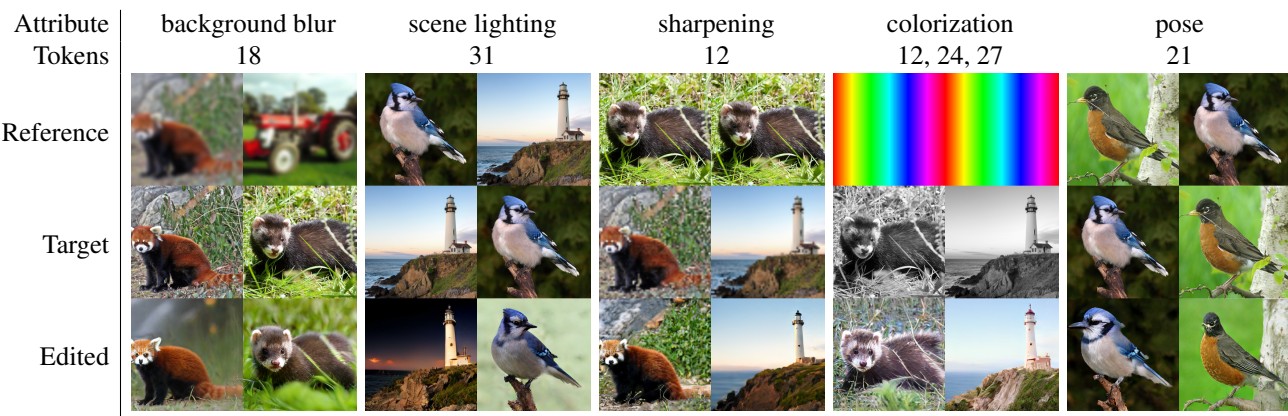

*Figure 3.* **Image editing via "copy and paste" latent space manipulation.** Edited image is obtained by replacing tokens in the target image with tokens copied from the reference image. Using the token positions previously identified to correspond to desired attributes, we can reliably obtain fine-grained control over appearance-related properties such as background blur, scene lighting conditions, and image quality. Class-specific semantic properties such as pose and gaze can also be controlled with some limited success (see last two columns for a successful and a failing example, respectively).

As shown in the rightmost two columns of Figure 2, many of the edits discovered with this single-token editing procedure produce valid images, and most of the edits are consistent with the results from the earlier class partition based token importance experiment.

### 3.3. "Copy & Paste" in Latent Space for Image Editing

Having discovered that certain high-level image properties appear to be represented by just one token position, and having demonstrated that perturbing tokens at these positions can produce valid images while affecting the desired high level properties, we propose an extremely simple "copy and paste" approach for direct latent space editing of images. Given an input image to edit $\mathcal{I}_{\text{target}}$ and a reference image $\mathcal{I}_{\text{ref}}$ that possesses an attribute that should be exhibited in the edited image $\mathcal{I}_{\text{edit}}$, we tokenize both the target and reference images with the 1D tokenizer and then simply copy the token lying at the position $k$ known to correspond to the desired attribute from the reference to the target tokens:

$$x_{\text{target}} = \text{Tok}(\mathcal{I}_{\text{target}}), \quad x_{\text{ref}} = \text{Tok}(\mathcal{I}_{\text{ref}}),$$
$$x_{\text{edit}} = \text{Replace}_k(x_{\text{target}}, x_{\text{ref}}^{(k)}), \quad (3)$$
$$\mathcal{I}_{\text{edit}} = \text{Dec}(x_{\text{edit}}).$$

Such a procedure cannot be expected to produce coherent results in the case of a 2D tokenizer, where single-token edits would result in spatially localized edits (Cao et al., 2023). Remarkably, in the case of 1D tokenization, it can produce interpretable, high-quality outputs such as those shown in Figure 3.

We observe that global appearance related properties — especially background blur, scene lighting and image quality/sharpness — can be accurately edited for essentially arbitrary target images. Furthermore, the reference image used to adjust these properties need not be semantically or visually related to the target image except in the appearance property to be transferred. In many cases, we can even use abstract synthetic reference images demonstrating the property to transfer. For example, we may use an arbitrary image blurred to the desired extent in order to precisely adjust the amount of background blur. We do not observe the same level of token compositionality for semantic properties. For example, transferring the pose of a subject by copying and replacing tokens is only successful to a limited extent (see last columns in Figure 3).

## 4. Gradient-Based Latent Optimization

In the preceding sections we have demonstrated that 1D tokenizers' highly compressed representation enables even extremely crude latent space manipulations to result in visually coherent and interpretable edits to input images. While this enables a surprising amount of control over certain pre-identified attributes (particularly those controlling global appearance), it is not particularly flexible, and cannot be reliably used for editing of more complex properties. In this section, we explore a more systematic and powerful image editing procedure based on gradient-based optimization of 1D tokens at test time.

### 4.1. Text-Guided Image Editing

We present a simple text-based image editing procedure based on test-time maximization of CLIP similarity (Radford et al., 2021) with respect to a user-supplied text prompt. First, the tokens to optimize are initialized from the input or *seed* image to be edited. These tokens are then iteratively

seed ———————— optimization ——————————→

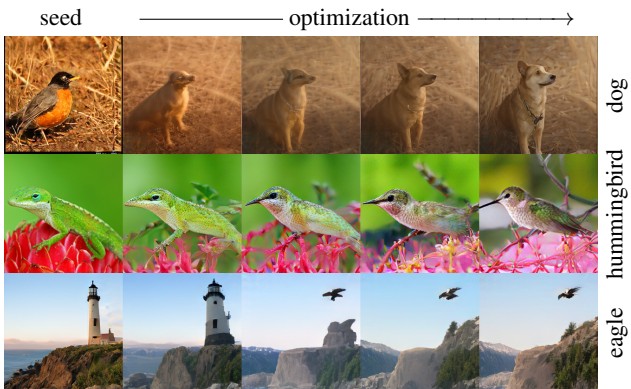

dog

hummingbird

eagle

*Figure 4.* **Test-time optimization according to a CLIP similarity objective.** Selected intermediate optimization results shown in middle columns, with final result after 400 optimizer iterations shown in the rightmost column. CLIP prompts are "a photo of a `class`", with `class` shown to the right.

updated by backpropagating the CLIP objective's gradient and performing gradient ascent as shown in Figure 1.

Specifically, we perform the following optimization:

$$\arg\max_{\hat{\mathbf{z}}^{(1)},...,\hat{\mathbf{z}}^{(K)}} \ell\left(\text{Dec}\left(\left[\text{VQ}_{\mathcal{D}}(\hat{\mathbf{z}}^{(1)}),...,\text{VQ}_{\mathcal{D}}(\hat{\mathbf{z}}^{(K)})\right]\right)\right),$$
(4)

where the decision variables are the continuous feature vectors $\hat{\mathbf{z}}^{(k)}$ as output by the tokenizer's encoder just prior to the vector quantization step $\mathbf{z}^{(k)} = \text{VQ}_{\mathcal{D}}(\hat{\mathbf{z}}^{(k)})$. The straight-through estimator is used for gradient computation through the vector quantization step. CLIP similarity of the decoded image with respect to the text prompt is denoted by $\ell$. Note that it is essential to include the vector quantization step within the optimization, as optimizing $\mathbf{z}^{(k)}$ directly leads to poor results. To solve the optimization in practice, we use the Adam optimizer with a learning rate of 0.1. A more detailed description of the algorithm can be found in Section C.1.

**A straightforward implementation of this optimization with no further tricks can already produce text-guided edits which are visually coherent and aligned with the prompt.** However, we do find that a few additional tweaks to smooth the CLIP objective's gradient, to artificially introduce additional stochasticity, and to add a small amount of regularization, can help. We ablate these design choices in Section 5.4.

Figure 4 shows examples of using our CLIP-guided token optimization to transform the main subject in an image into a different class. However, the usage of CLIP guidance allows going beyond class-conditional generation. We demonstrate the capability for open-domain text-guided editing of a subject's context in Figure 5, noting that the optimization preserves key aspects of the subject while aligning the generated image with the prompt.

input image     edit 1     edit 2     edit 3

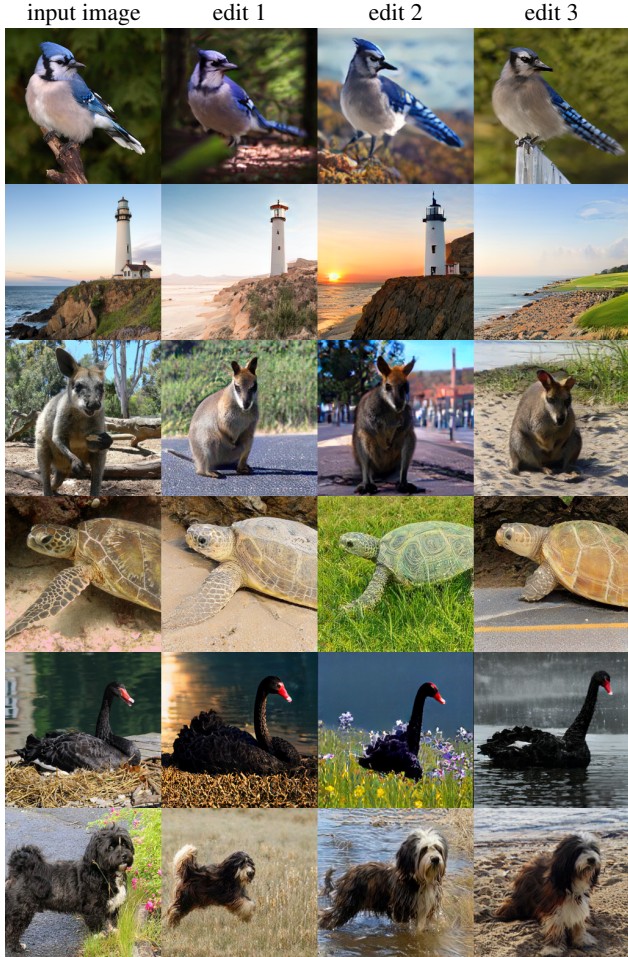

*Figure 5.* **Flexible text-guided editing of an input image using test-time optimization with CLIP gradients.** Key attributes of the input image such as pose are preserved while aligning the image with the prompt. Prompts are listed in the appendix (Table A1).

### 4.2. Text-to-Image Generation

Consider now the case in which no seed image is available. In this case, we can apply the test-time optimization procedure starting with randomly selected tokens. In particular, we initialize the feature vectors $\hat{\mathbf{z}}^{(k)}$ (prior to the vector quantization step) by randomly sampling their value from a normal distribution. We find that this works better than initializing from tokens randomly selected from the codebook, and allows for successful **text-to-image generation without the need for a seed image** (see Figure 6 for example generations). A further discussion of tweaks and parameters used for this "from-scratch" text-to-image generation task can be found in Section 5.5

### 4.3. Inpainting

Test-time optimization of tokens provides a general framework for optimizing arbitrary cost functions, and is not

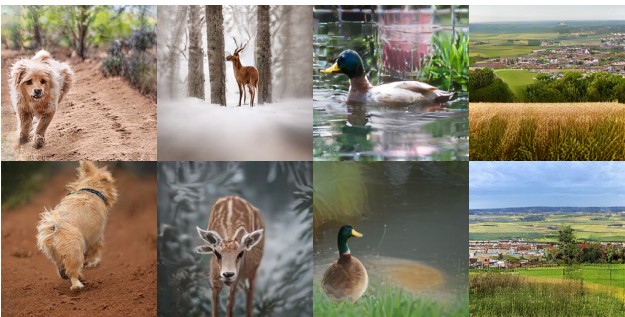

*Figure 6.* **"From-scratch" generation starting from random tokens.** Instead of starting with an initial image, it is also possible to apply our test-time latent token optimization starting from a randomly sampled initialization. Selected samples for prompts "a dog running on a dirt trail", "majestic deer standing in a snowy forest", "a duck in a pond," and "a wide landscape vista overlooking fields and a town".

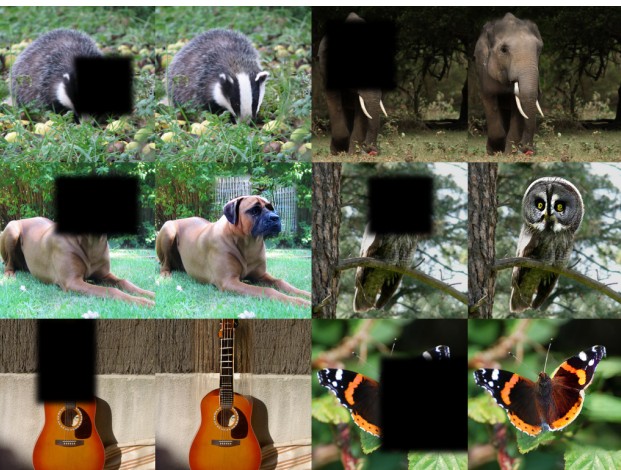

*Figure 7.* **Inpainting via test-time optimization of tokens.** For each pair of images, the left image is the input to the optimization procedure, which optimizes 1D tokens according to a reconstruction loss on the given part of the image. The final image is produced by blending the input with the reconstruction.

limited to CLIP similarity scores. As an additional use case, we consider the inpainting problem. In this case, the test-time optimization is modified to use an L1 reconstruction loss over the unmasked part of the image.

While this modified algorithm does perform the inpainting task successfully in some cases, we observe that the reconstruction loss tends to favor blurry reconstructions. To solve this issue, we periodically reset the tokens during the optimization by (i) decoding them into an image, (ii) replacing the parts of the image which are known with their given values, and (iii) re-encoding this modified image to obtain the reset tokens. We also find it essential to inject noise during the optimization process and to apply L2 regularization on the tokens. Example results are visualized in Figure 7.

# 5. Evaluation

For quantitative evaluation of editing and generation quality, we will consider a class-conditional generation pipeline based on a small *seed image* dataset subsampled from the ImageNet training data, along with a set of CLIP text prompts used to guide generation towards target classes. This setting is similar to semi-parametric generation such as instance conditioning (Casanova et al., 2021) or retrieval-augmented diffusion (Blattmann et al., 2022).

**Seed image set.** A fixed number of ImageNet ILSVRC2012 (Deng et al., 2009) training set images are randomly selected. For each ImageNet class, an equal number of images is sampled at random without replacement. The 1D tokens for images from this small *seed image dataset* are used to initialize the test-time token optimization.

**CLIP prompts.** Prompts of the form "a photo of a `class`" are constructed. We generate 50k prompts, distributed according to the ImageNet validation set class statistics. In each prompt, `class` is the natural language label for an ImageNet synset (the first WordNet lemma name).

**Class-to-seed association.** We consider a number of different strategies to associate a target prompt with a seed image. First, we consider random association. Second, we can use CLIP similarity to associate each prompt with either the most or one of the (randomly picked) top-k most similar seed images.

**Models.** Our evaluation uses the pretrained TiTok tokenizer checkpoints[1] released by the TiTok authors. We refer to the models by name as VQ|VAE-$S_{enc}S_{dec}$-$K$ where the prefix stands for the type of autoencoder (VQ: discrete tokens using vector quantization, VAE: continuous tokens with KL-regularization), $S_{enc}$ and $S_{dec}$ refer to the size (L, B, or S) of encoder and decoder, respectively, and $K$ denotes the number of tokens.

**Computational cost.** Running 300 iterations of the text-guided image editing optimization (with CLIP loss smoothed over 8 random crops) in half precision using the VQ-LL-32 tokenizer takes 7 seconds per image on an NVIDIA A100. Most experiments in this section use this configuration.

## 5.1. Evaluation Protocol

In order to evaluate quality of generations, we compute the Fréchet Inception Distance (FID) (Heusel et al., 2017) with respect to the ImageNet training statistics following the protocol from ADM (Dhariwal & Nichol, 2021). We also report Inception Score (IS) (Salimans et al., 2016).

---

[1] https://github.com/bytedance/
1d-tokenizer/blob/main/README_TiTok.md

*Table 1.* **A limited number of seed images combined with diverse prompts can achieve good FID and prompt alignment.** *Baseline* refers to directly using the seed images (reconstructed via TiTok VQ-LL-32) without any optimization. *Adversarial* refers to direct optimization of pixels to maximize a CLIP objective intended to demonstrate the immunity of the SigLIP similarity. *Num Seeds* denotes the size of the pool of randomly selected seed images, while *Seed Assoc* describes whether seeds are randomly assigned to target prompts or use CLIP-based selection. Configuration marked (∗) used for remaining experiments in this section.

| | Num Seeds | Seed Assoc | Method | FID-5k↓ | | IS↑ | | CLIP↑ | | SigLIP↑ | |
|---|---|---|---|---|---|---|---|---|---|---|---|
| | 2000 | random | baseline | 25.2 | | 293 | | 0.12 | | -16.2 | |
| | | | adversarial | 262 | (+241) | 2 | (-291) | 0.59 | (+0.47) | -14.4 | (+1.8) |
| | | | VQ-LL-32 w/ token opt. | 20.7 | (-4.5) | 75 | (-218) | 0.36 | (+0.24) | -0.3 | (+15.9) |
| | 2000 | CLIP top-1% | baseline | 46.1 | | 226 | | 0.25 | | -6.9 | |
| | | | VQ-LL-32 w/ token opt. | 14.6 | (-31.5) | 161 | (-65) | 0.39 | (+0.15) | 2.6 | (+9.6) |
| (∗) | 1000 | CLIP top-1% | baseline | 73.7 | | 197 | | 0.24 | | -7.0 | |
| | | | VQ-LL-32 w/ token opt. | 15.1 | (-58.6) | 160 | (-37) | 0.39 | (+0.15) | 2.5 | (+9.5) |
| | 1000 | CLIP top-1 | baseline | 71.9 | | 243 | | 0.31 | | -1.3 | |
| | | | VQ-LL-32 w/ token opt. | 21.2 | (-50.7) | 281 | (+38) | 0.40 | (+0.09) | 3.5 | (+4.8) |
| | 500 | CLIP top-1% | baseline | 119.2 | | 135 | | 0.25 | | -7.0 | |
| | | | VQ-LL-32 w/ token opt. | 16.6 | (-102.6) | 155 | (+20) | 0.39 | (+0.14) | 2.5 | (+9.5) |

To evaluate alignment with the target prompt, we consider CLIP as well as SigLIP (Zhai et al., 2023) similarity. It is critical to include SigLIP scores, as the CLIP-guided test-time optimization resembles an adversarial attack on the CLIP model and a high CLIP similarity therefore does not necessarily indicate successful alignment of the generated image with the prompt (see Table 1 for an example of maximizing CLIP score without significantly improving SigLIP score).

## 5.2. Seed Image Set Size and Association

To assess the degree of the test-time optimization's reliance on seed images as a source of diversity, we vary the size of the seed image set. We further check the optimization's ability to handle diverse seed images with varying degrees of similarity to the target prompt by ablating the CLIP-based seed association, and vary the degree of stochasticity in the prompt-to-seed association by selecting either the most similar seed image or picking uniformly at random between the top 1% of most similar seed images.

**We find that only a small number of seed images (less than 1000) is needed to produce diverse generations, achieving an FID of 8.6 for 50k samples.** We also find that the number of seed-to-prompt combinations is an important source of diversity, observing that seed association schemes that favor more diversity in the association (i.e., allowing for associations that do not necessarily have the best similarity) outperform those favoring higher similarity of seed and prompt in terms of FID, although those with less diversity can lead to improved IS. Varying the degree of stochasticity in the seed selection therefore provides a way to control the tradeoff between diversity and quality. Performing random

*Table 2.* **Increased compression leads to improved generation.** Usage of tokenizers with increasing number of tokens or increasing codebook size $|\mathcal{D}|$ degrades generative performance. The number of optimization iterations used corresponds to the best FID-5k score achieved.

| Tokenizer | $|\mathcal{D}|$ | FID-5k↓ | IS↑ | CLIP↑ | SigLIP↑ |
|---|---|---|---|---|---|
| VQ-LL-32 | 4096 | 15.1 | 160 | 0.39 | 2.53 |
| VQ-BB-64 | 4096 | 16.3 | 135 | 0.40 | 2.77 |
| VQ-BL-64 | 8192 | 18.6 | 106 | 0.40 | 2.07 |
| VQ-BL-128 | 8192 | 22.3 | 92 | 0.42 | 2.39 |

association leads to poor results, indicating that the test time optimization struggles to generate images when the seed image is unrelated to the prompt. Table 1 summarizes these results.

**Note on determinism.** The experiments in this section include a small amount of randomness in the loss. This is due to nondeterminism in some of the CUDA kernels used in our PyTorch implementation. Interestingly, this is sufficient to lead to diverse generations by the test-time optimization procedure. Note that in the case of at least 1000 seed images with top-1% selection, there is enough diversity in the seed-to-prompt association to yield about 4000 unique inputs, such that the FID-5k results would be relatively unaffected by the use of a fully deterministic implementation.

## 5.3. Tokenizer Variations

**Generative performance improves with increasing compression by the tokenizer.** As shown in Table 2, a decreasing number of tokens as well as a decreasing codebook size lead to significant improvements generation quality. Out of

*Table 3.* **Vector quantization and compression enabled by 1D tokenization are key to generative performance.** Test-time optimization using 1D continuous tokens or 2D discrete tokens fails. In case of continuous tokens, we apply L2 regularization with weight chosen for best results from a parameter sweep. Note that when using the tokenizer trained with VQ but optimizing without VQ (second row), performance drops significantly.

| Tokenizer | FID-5k↓ | IS↑ | CLIP↑ | SigLIP↑ |
|---|---|---|---|---|
| VQ-LL-32 | 15.1 | 160 | 0.39 | 2.53 |
| VQ-LL-32 *w/o VQ* | 17.1 | 127 | 0.43 | 3.05 |
| VAE-LL-32 | 33.2 | 93 | 0.46 | 3.05 |
| MaskGIT-VQGAN | 34.3 | 73 | 0.44 | 2.42 |

*Table 4.* **Ablation study.** All using the VQ-LL-32 tokenizer and 300 optimizer iterations. All metrics other than FID-50k computed over 5000 samples. We use the configuration from the line marked (∗) for the preceding experiments in this section, unless noted otherwise.

| | FID 5k (50k) ↓ | | IS↑ | CLIP↑ | SigLIP↑ |
|---|---|---|---|---|---|
| baseline | 14.8 | | 140 | 0.39 | 1.81 |
| +random crops | 15.3 | | 157 | 0.40 | 2.55 |
| (∗) +EMA | 15.1 | (8.6) | 160 | 0.39 | 2.53 |
| (−100 opt. iter) | 15.2 | | 149 | 0.38 | 1.93 |
| (+100 opt. iter) | 15.3 | | *169* | 0.40 | 2.88 |
| +token noise | **14.5** | **(8.2)** | 165 | 0.39 | 2.46 |
| +token reg. | *14.6* | **(8.2)** | **182** | 0.38 | 2.07 |

all models we tested, were only able to achieve qualitatively satisfactory performance in editing and generation tasks using the TiTok-LL-32 model (see Figure A6 in the appendix for qualitative results).

**Discrete vs. continuous tokens.** In comparing a version of the TiTok tokenizer trained as a VAE with continuous tokens with the standard VQ model, we find that the *compression provided by the discrete latent space is essential* in achieving good generative performance (Table 3). Even when optimizing the continuous tokens of a tokenizer trained with VQ (i.e. optimizing $\mathbf{z}^{(k)}$ directly and skipping the VQ step), performance drops significantly. We hypothesize that VQ provides an essential form of regularization that prevents the optimization from behaving adversarially (Santurkar et al., 2019).

**1D vs. 2D tokenizer.** We further find that the large number of tokens in a standard VQGAN 2D tokenizer such as MaskGIT's VQGAN (Chang et al., 2022) prevent successful application of the test-time optimization approach for generation (Table 3). This is consistent with the finding that an increasing number of tokens is detrimental to generation (Table 2). Note also that the 32-token 1D tokenizer is able to successfully generate visually coherent images even without any tricks such as augmentations to stabilize the

*Table 5.* **Generating images from scratch.** It is possible to generate images even when starting from randomly initialized tokens as opposed to seed images. When using our vanilla token optimization algorithm, we observe a decrease in generation quality. However, additional optimization iterations as well as the tweaks presented in Section 5.4 help to significantly decrease the gap in generation performance when comparing seed-image-based initialization and random token initialization.

| Tweaks? | Seed | Iter. | FID 5k↓ | IS↑ | CLIP↑ | SigLIP↑ |
|---|---|---|---|---|---|---|
| no | CLIP-top-1% | 300 | 15.1 | 160 | 0.39 | 2.53 |
| | **random tokens** | 300 | 17.0 | 114 | 0.38 | 1.68 |
| yes | CLIP-top-1% | 300 | 14.6 | 182 | 0.38 | 2.07 |
| | **random tokens** | 400 | 15.5 | 152 | 0.40 | 2.54 |

loss function, while the 2D tokenizers' spatially arranged tokens tend to lead to optimization results that are spatially incoherent.

**Number of optimization iterations.** It is possible to control the tradeoff between FID and IS scores by varying the number of optimization iterations. In fact, the best FID and best IS scores are achieved at different numbers of iterations, and this number varies between different tokenizers. Therefore, we report values at the number of iterations yielding the best FID score in Table 2, and include additional results in Section C.2.

### 5.4. Tweaking the Test Time Optimization Algorithm

We find that averaging the CLIP loss over multiple random crops (we use 8 crops encompassing 75% of the image area) significantly helps prompt alignment and boosts IS, while slightly degrading FID, likely due to smoother gradients of the objective function leading to reduced diversity in the generations. Returning an exponential moving average (EMA) of token iterates yields a small boost to FID and IS. Injecting additional noise to the tokens at the beginning of each optimization iteration can further improve FID and IS, achieving an FID of 8.2 over 50k samples. Applying weak L2 regularization on the tokens yields a significant boost to IS, while slightly degrading FID. Finally, increasing the number of optimization iterations can improve IS as well as CLIP and SigLIP scores at the expense of degraded FID. Table 4 summarizes these findings. Additional results to establish the relationship between optimization iterations and FID/IS are presented in the appendix (Section C.2).

While these tweaks do improve performance, we note that our very simple baseline is already strong. This highlights that the pretrained **tokenizer on its own possesses strong generative capabilities** and can easily be made to generate images with simple heuristic methods.

*Table 6.* **System-level comparison.** Approaches other than our own require training a generative model (gray). We compare with models supporting unconditional generation, as our method supports arbitrary objective functions and the 1D tokenizer is trained in a self-supervised manner without class labels. We also compare with *semi-parametric* methods leveraging retrieval of training set images. The second column specifies whether arbitrary objective functions can be used as guidance without training a new model. The third column refers to the need to access training set images at test time, where *selected* denotes that the entire training set has been considered in selection of the pool of training images available at test time, while *random* describes a randomly subsampled pool of seed images.

| Gen. Model Training | Plug & Play Guidance | Access to Training Data | Method | FID↓ | IS↑ |
|---|---|---|---|---|---|
| Yes | No | No | ADDP (Tian et al., 2024) | 7.6 | 105 |
| | | | RCG-G (Li et al., 2024a) | 2.2 | 254 |
| Yes | No | Full training set | RCDM (Bordes et al., 2022) | 19.0 | 52 |
| | | 1000 selected images | IC-GAN (Casanova et al., 2021) | 15.6 | 59 |
| | | Full training set | RDM-IN (Blattmann et al., 2022) | 5.9 | 159 |
| **No** | **Yes** | 1000 random images | **Test-time optimization** (VQ-LL-32) | 8.2 | 182 |

### 5.5. Text-to-Image Generation Without Seed Image

When no seed image is available, it is still possible to apply the test-time optimization approach for text-to-image generation. As reported in Table 5, FID and IS scores are weaker than when starting with a seed image. While this is expected, the tweaks from Section 5.4 as well as additional optimization iterations help significantly close the gap in performance. Note that image generation quality when starting from randomly selected tokens (that is, $\hat{\mathbf{z}}^{(k)} \sim \mathcal{N}(0, \sigma_{\text{init}}^2)$) is *higher* than when starting from random but valid seed images (see third row in Table 1). In our experiments, we set $\sigma_{\text{init}} = 0.3$ as chosen for best performance from a sweep including $\{0.05, 0.3, 1.0\}$.

### 5.6. System-Level Comparison

Since tokenizers are trained unconditionally, we focus our system-level comparison in Table 6 on methods supporting unconditional generation. We find that even though image generation via test-time optimization of tokens is training-free, it approaches or exceeds performance of previous unconditional generation approaches evaluated on ImageNet (Tian et al., 2024), including semi-parametric ones which also require access to a database of images (Bordes et al., 2022; Casanova et al., 2021; Blattmann et al., 2022). Compared to these generative models, our tokenizer-only approach can maintain high diversity of generations with even smaller seed dataset sizes.

Recent unconditional generation methods such as RCG (Li et al., 2024a) can match or exceed the performance of class-conditioned generators. However, we stress that RCG and all other approaches we compare against require training of a generative model. **Overall, we conclude that the pretrained TiTok VQ-LL-32 tokenizer is able to achieve reasonable generation quality without any further training** through the use of a straightforward test-time optimization approach.

Qualitative image generation results can be found in the appendix (Figure A9).

## 6. Discussion

1D tokenizers have recently shown impressive compression of images into small numbers of discrete tokens. Compared to generative models using 2D tokenizers, models trained in this highly compressed latent space produce high quality images orders of magnitude faster without significantly compromising sample quality. In our paper, we show that this is facilitated by the strong generative capabilities of the tokenizer itself. We find that the latent space of 1D tokens is highly semantic, to the point that certain image editing operations can be performed in an accurate and interpretable way through targeted perturbations of specific tokens. Furthermore, a 1D tokenizer can generate diverse and high quality images without the use of any dedicated generative model at all with assistance from a simple test-time optimization procedure.

Our experiments suggest that increasing compression — in the form of a small number of tokens, vector quantization, and small codebook size — is crucial in giving rise to generative capabilities in the tokenizer. We therefore hope our work encourages scaling of tokenizers to even higher compression ratios, larger datasets, or different domains.

## Impact Statement

This paper presents work that is intended to improve understanding of image tokenizers with very high compression ratios. It does not advance the state of the art in image generation, although it may encourage work that does. Improved image understanding and generation have many potential societal consequences, none which we feel must be specifically highlighted here.

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

# A. Additional Background on TiTok Tokenizer

For completeness, we provide a graphical overview of the TiTok 1D tokenizer in Figure A1. 2D tokens are first obtained by patchifying the input image and embedding the patches, which interact with the 1D tokens via several layers of bidirectional attention. TiTok relies on an off-the-shelf 2D tokenizer (MaskGIT's VQGAN) to reconstruct an image from the set of 2D tokens predicted by the decoder, and is trained in two stages. First, the objective is to learn to reconstruct the (frozen) VQGAN's tokens, and later, the VQGAN itself is also fine-tuned (Yu et al., 2024b).

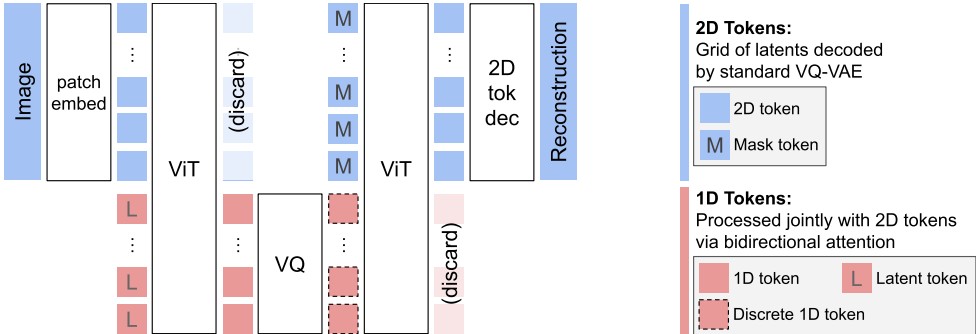

*Figure A1.* **Architecture of the TiTok (Yu et al., 2024b) 1D tokenizer.** An image is encoded into a short sequence (e.g. 32 tokens) of "1D" tokens, which unlike the grid-arranged latents from a standard 2D tokenizer, do not follow any particular spatial structure and are therefore able to capture global attributes of the image.

# B. Token Semantics

In Section 3.1 we introduce a metric for token importance based on the variance of averaged features across different partitions of the ImageNet validation set. In the following, we first provide additional experiments to further support the effectiveness of this metric in capturing token importance (Section B.1). Then, we present additional results showing token importance for different tokenizers (Section B.2).

## B.1. Linear Probing

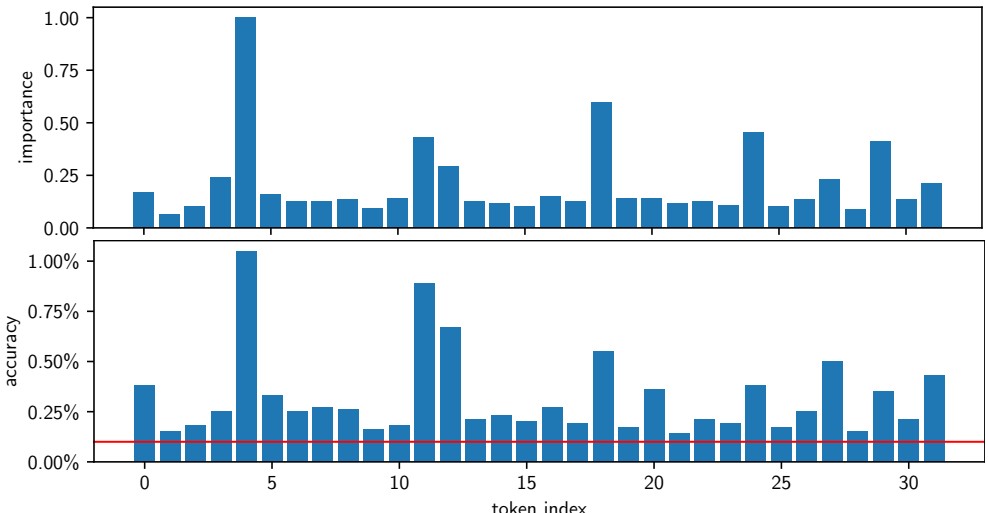

*Figure A2.* **ImageNet classification accuracy using individual discrete tokens is poor** (bottom), **but correlated with the proposed importance metric** (top). Red line at 0.1% corresponds to random guessing among the 1000 possible classes. Importance metric (top) is computed according to 3.1.

**Classification using only one token position at a time.** In Figure A2 we attempt ImageNet-1k classification using discrete tokens, one-at-a-time. Since we here only consider a single token at a time, we can directly learn a table mapping token

value to class logits for each token position. We observe strong correlation between the results of this experiment and the importance metric we propose in Section 3.1.

**Linear probing with iterative token masking.** Next, we perform linear probing on multiple tokens at a time, and consider both the high-dimensional transformer features from the last layer of TiTok's encoder, as well as discrete tokens as in the previous experiment. In contrast to the previous experiment, we begin by linear probing all 32 available tokens, and iteratively remove the "least important" ones. Our exact protocol for linear probing with iterative token removal is as follows:

1. Start by training a linear head on top of the concatenation of all 32 features.

2. During training, perform dropout to randomly mask features corresponding to certain token positions.

3. After a fixed number of epochs, evaluate the model on the IN-1k validation data with no dropout.

4. Select the most "unimportant" by applying dropout one-by-one at each token position. After running several validations with each remaining token position masked, select the token position which leads to the smallest drop in classification accuracy with respect to the previous validation results that do not apply masking.

5. Repeat this process from the start (1), until all token positions are masked.

In the case of using discrete token indices, we learn a new codebook of features to perform linear probing on. Results can be found in Figure A3.

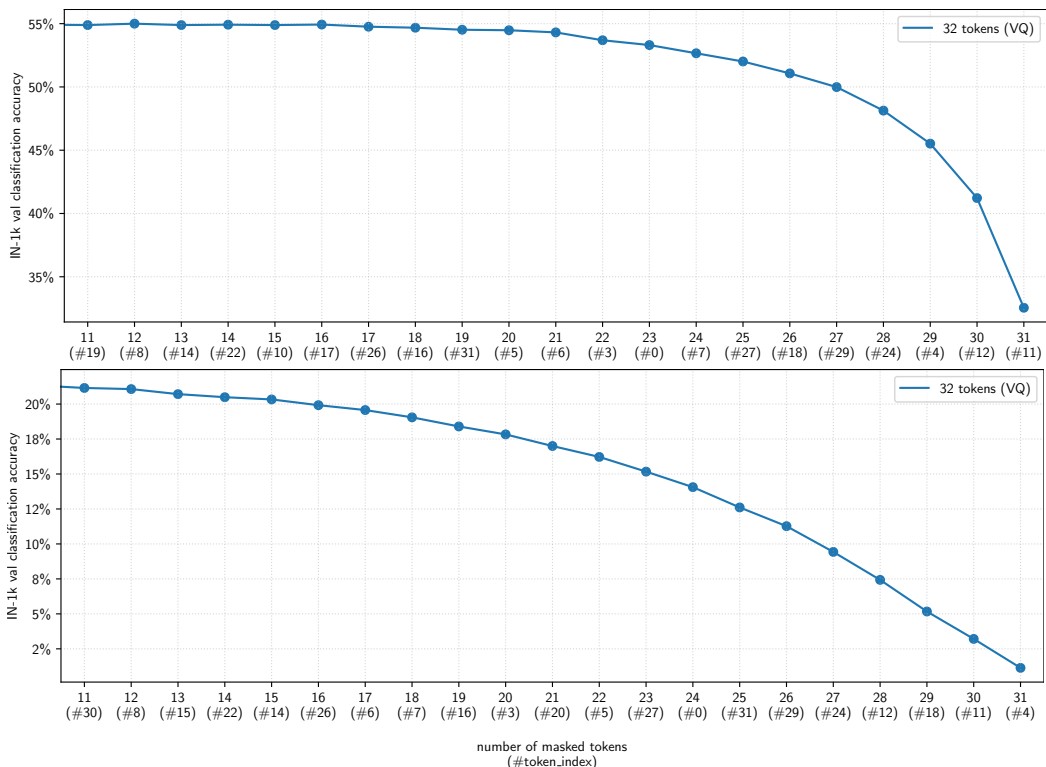

*Figure A3.* **Results from linear probing while iteratively removing tokens with the smallest impact on classification accuracy largely match our token importance metric.** Top: probing of high dimensional features produced by last transformer layer in TiTok encoder. Bottom: linear probing after vector quantization, using discrete token indices. In both cases, we plot IN-1k classification accuracy as a function of number of masked tokens. Token index to remove after each masking iteration is shown in parentheses, so that most important token positions can be read off right-to-left. For example, the top plot based on high-dimensional features suggests tokens 11, 12, 4, 24, ... as most important, while the discrete-token-based plot on the bottom indicates most important tokens are 4, 11, 18, 12, ....

Results from these experiments are largely aligned with those from the token importance shown in Figure A2, indicating that the approach suggested in Section 3.1 is indeed an informative token importance computation.

## B.2. Per-Token Inter-Class Variance and Linear Probing for VAE-TiTok

In Figure A4 we repeat the token importance experiment from Section 3.1 for additional class partitions. Furthermore, we find that the VAE version of TiTok does not exhibit the same high degree of correspondence between semantics and token position.

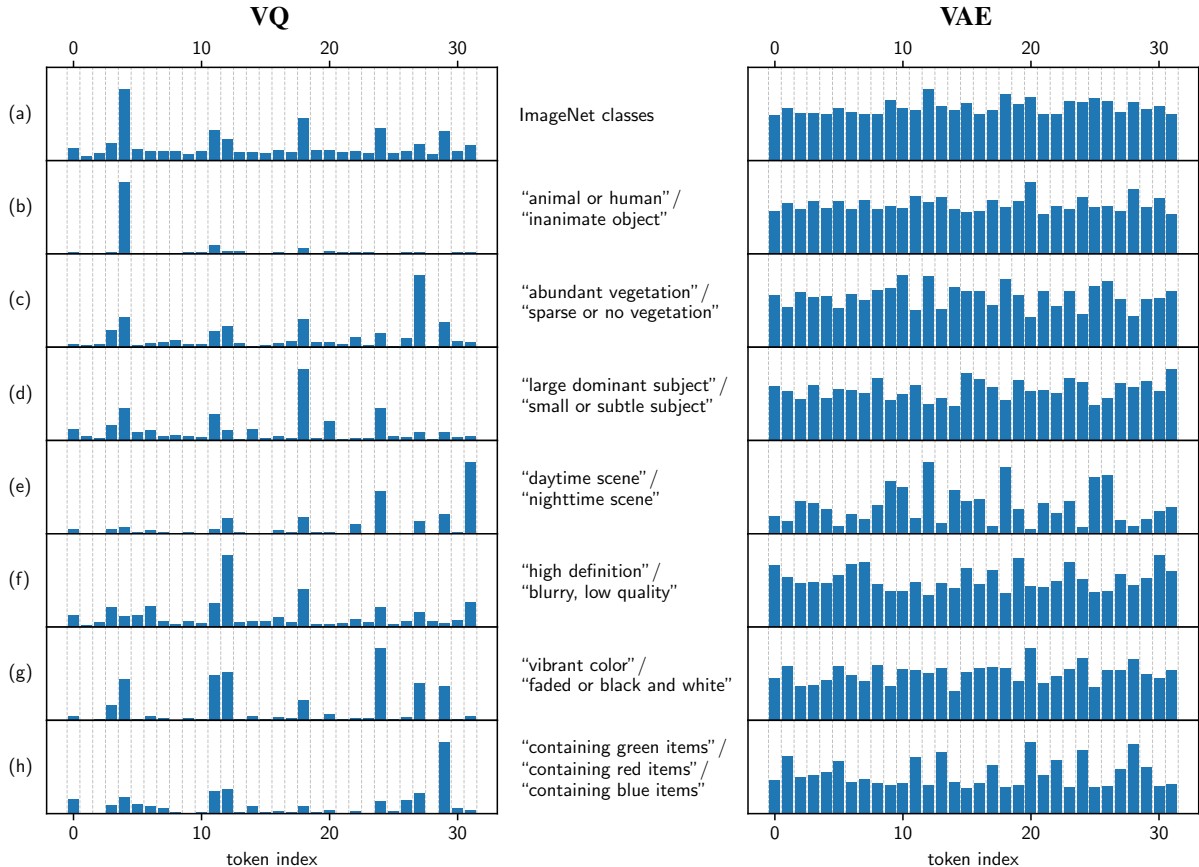

*Figure A4.* **VAE-TiTok does not show semantic disentanglement across token positions.** We plot token importance computed according to Section 3.1 for both the VQ (left) and VAE (right) versions of the LL-32 TiTok tokenizer for various class partitions of the ImageNet validation set. Normalized token importance is shown on the y axis. Abbreviated prompts used as labels for CLIP-based class assignment shown in the middle.

# C. Gradient-Based Latent Editing

## C.1. Detailed Description of Test-Time Optimization Algorithm

The following algorithm describes the baseline algorithm for text-guided image editing.

---

**Algorithm A1** Test-Time Optimization For CLIP-Guided Latent Editing.

---

**Input:** `img` the seed image, and `prompt` a text prompt
**Output:** `recons` the optimized image
1: `tokens` $\leftarrow$ TiTokEnc(`img`)
2: `prompt_enc` $\leftarrow$ CLIPTextEnc(`prompt`)
3: **loop**
4:     `recons` $\leftarrow$ (TiTokDec $\circ$ TiTokQuant)(`tokens`)
5:     `rec_enc` $\leftarrow$ CLIPImageEnc(`recons`)
6:     Gradient ascent step on $\nabla_{\text{tokens}} \frac{\text{rec\_enc}^{\top}\text{prompt\_enc}}{\|\text{rec\_enc}\|\|\text{prompt\_enc}\|}$
7: **end loop**

---

In Algorithm A1, we refer to TiTok's transformer-based image encoder taking images and producing $K$ $D$-dimensional continuous features as TiTokEnc $: [0,1]^{C\times H\times W} \rightarrow \mathbb{R}^{K\times D}$. The continuous features are then quantized to a particular entry in the codebook via the vector quantization module TiTokQuant $: \mathbb{R}^{K\times D} \rightarrow \mathbb{R}^{K\times D}$, using straight-through gradient estimates for the backward pass. Quantized features are decoded to an image via the decoder TiTokDec $: \mathbb{R}^{K\times D} \rightarrow [0,1]^{C\times H\times W}$. In practice, we use the Adam optimizer with a learning rate of $0.1$, $\beta_1 = 0.9$ and $\beta_2 = 0.999$.

Additional changes we use for the inpainting task as well as in the improved versions from Table 4 can be found below.

---

**Algorithm A2** Test-Time Optimization with Optional Tweaks.

---

**Input:** `img` the seed image, $\ell$ an objective function taking an image
**Output:** `recons` the optimized image
1: `tokens` $\leftarrow$ TiTokEnc(`img`)
2: **for** $i \leftarrow 1$ to $N_{\text{iter}}$ **do**
3:     sample `noise` $\sim \mathcal{N}(\mathbf{0}, \sigma_i^2\mathbf{I})$
4:     `tokens` $\leftarrow$ `tokens` + `noise`
5:     `recons` $\leftarrow$ (TiTokDec $\circ$ TiTokQuant)(`tokens`)
6:     Gradient ascent step on $\nabla_{\text{tokens}} \left( \ell(\text{recons}) + \lambda\frac{1}{K}\sum_{z^{(k)}\in\text{tokens}} \left\| z^k \right\|_2^2 \right)$
7:     `tokens` $\leftarrow$ TiTokEnc(Reset(`recons`))
8: **end for**

---

Algorithm A2 shows addition of token noise in blue. We use a cosine schedule to ramp the noise from $\sigma_1^2 = 0.3$ to $\sigma_{200}^2 = 0$. Token regularization is highlighted in green. We obtain best results with $\lambda = 0.02$. Token EMA (not shown) uses a decay factor of $0.98$.

**Inpainting.** The token reset procedure highlighted in red is only used for the inpainting task. Here, Reset takes the decoded image and blends it with the given parts of the image according to the inpainting mask. For inpainting, we find it important to use masks with soft transitions from fully masked to fully unmasked. Such a "soft" mask can be obtained, for example, by applying Gaussian blur to a binary mask. The reasons we use soft masks are twofold: first, it allows for graceful blending of the unmasked (given) part of the image with the generated image, which may appear discontinuous if blended with a binary mask that uses sharp edges. Second, we find that the optimization can be reliably initialized with masked images when the mask has a soft transition (such as those in Figure 7), however, the optimization struggles to succeed when initialized with images containing sharp edges.

## C.2. Choosing Optimization Iterations

The number of optimization iterations directly impacts the quality of generations. Intuitively, the longer the optimization is run, the further the image will deviate from the seed image. This can be desirable or necessary when the seed image needs significant modification in order to match the target prompt, but for "smaller" edits, a lesser number of iterations may be chosen. Therefore, the number of iterations should in general be picked on an example-by-example basis, and one could design adaptive stopping criteria based on monitoring the objective function's value and change across iterations. We leave this to future work.

For the evaluations performed in Section 5, we report either scores for a fixed number of optimization iterations, or scores corresponding to the number of iterations yielding the best FID. For completeness, we provide FID and IS for a sweep of optimization iterations in Figure A5 below.

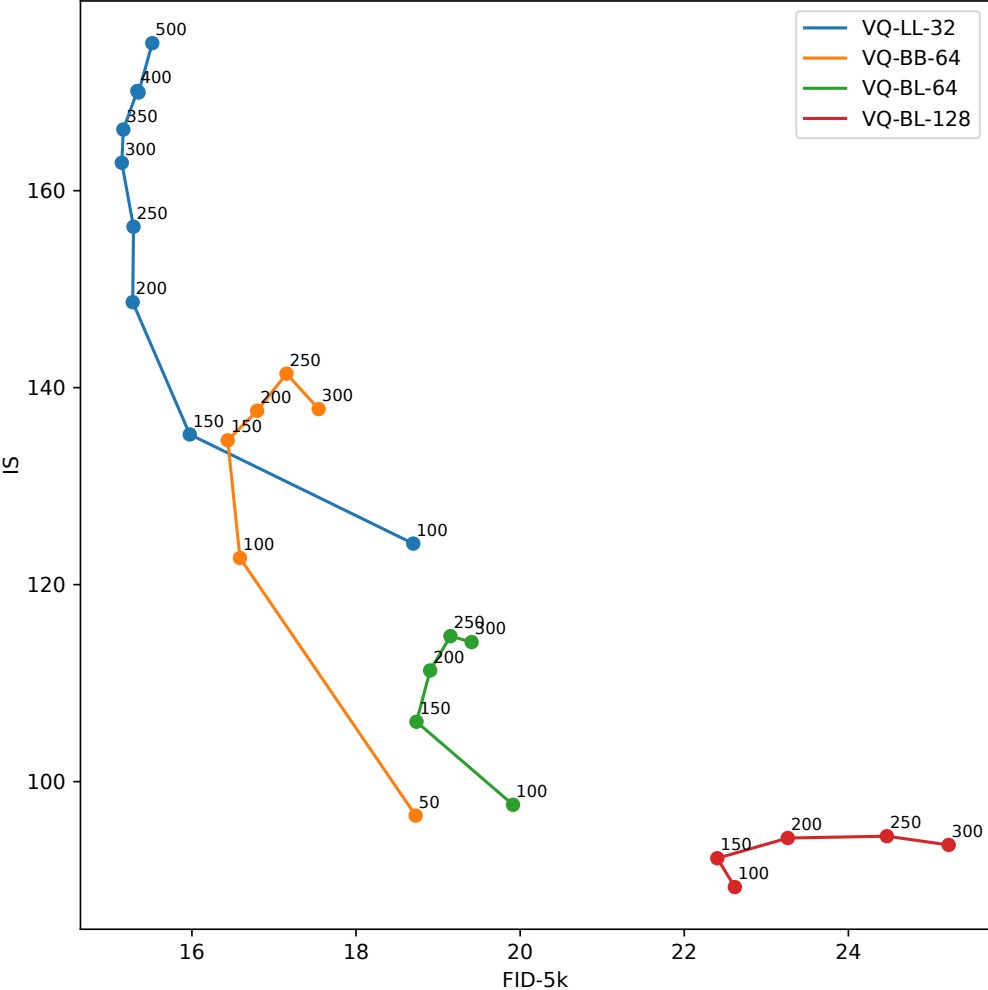

*Figure A5.* **The number of optimization iterations controls the tradeoff between FID and IS.** The small number alongside each point denotes the number of optimization iterations used. The optimal number of iterations for either FID, IS or a given FID/IS tradeoff is dependent on the tokenizer used, and models with higher compression ratio are shown to require a larger number of iterations to achieve peak performance.

## C.3. Image Editing Example Prompts

*Table A1.* Prompts used for Figure 5

| Prompt Prefix | Edit 1 | Edit 2 | Edit 3 |
|---|---|---|---|
| a photo of a blue jay... | ...in a forest | ...with the ocean in the background | ...perched on a fence |
| a photo of a lighthouse... | ...in the desert | ...at sunset | a golf course on the coast |
| a photo of a wallaby... | ...on the highway | ...in a town | ...on the beach |
| a photo of a turtle... | ...on the beach | ...in a green field | ...on a road |
| a photo of a black swan... | ...at sunset | ...among wildflowers | ...in the rain |
| a photo of a tibetan terrier... | ...running in a field | ...wading in a river | ...on the beach |

# D. Additional Qualitative Results

We include additional qualitative results on the importance of compression (Figure A6) and on the out-of-domain tasks of generating images of classes not present in ImageNet (Figure A7) as well as style transfer to non-natural-image styles (Figure A8). Finally, we show selected generations of the best configuration (32-token tokenizer with VQ) alongside their seed images and prompts in Figure A9.

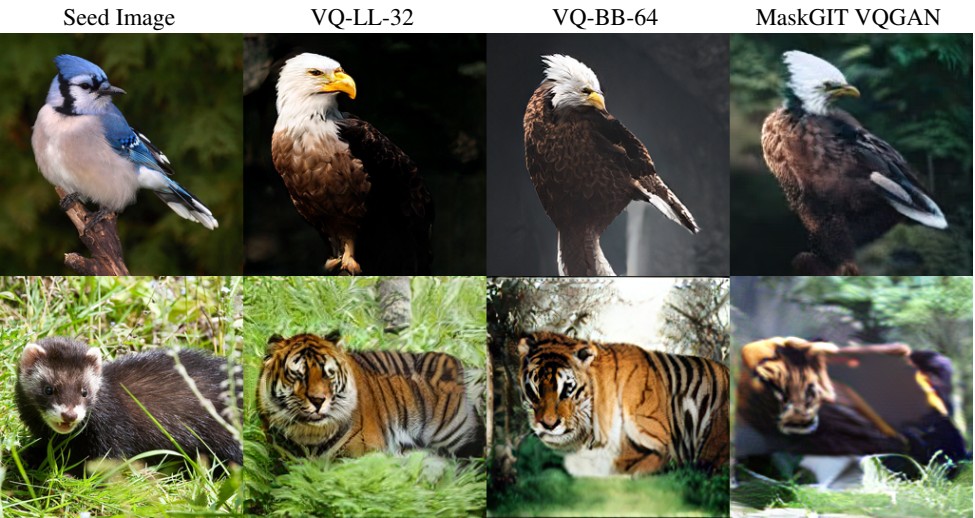

*Figure A6.* **Qualitatively, the gap in generation quality is very significant between the TiTok 32-token, 64-token, and VQGAN models.** Prompts are "a photo of an eagle" and "a photo of a tiger" for the first and second rows, respectively.

Cardinal   Capybara   Rhinoceros

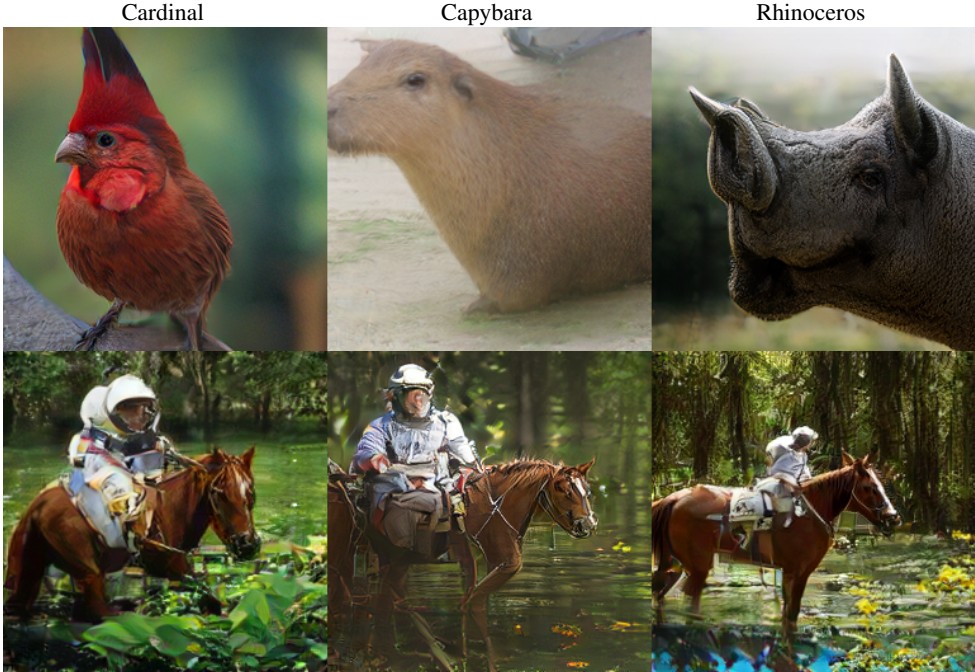

A photo of an astronaut riding a horse in the forest. There is a river in front of them with water lilies.

*Figure A7.* **Generating images of animals not included in the ImageNet dataset (top) or open-domain scenarios including subjects not included in ImageNet (bottom)** can produce inaccurate results with varying degrees of plausibility, since the tokenizer has only been trained on ImageNet.

input image   a watercolor painting of a lighthouse   a comic-style drawing of a lighthouse

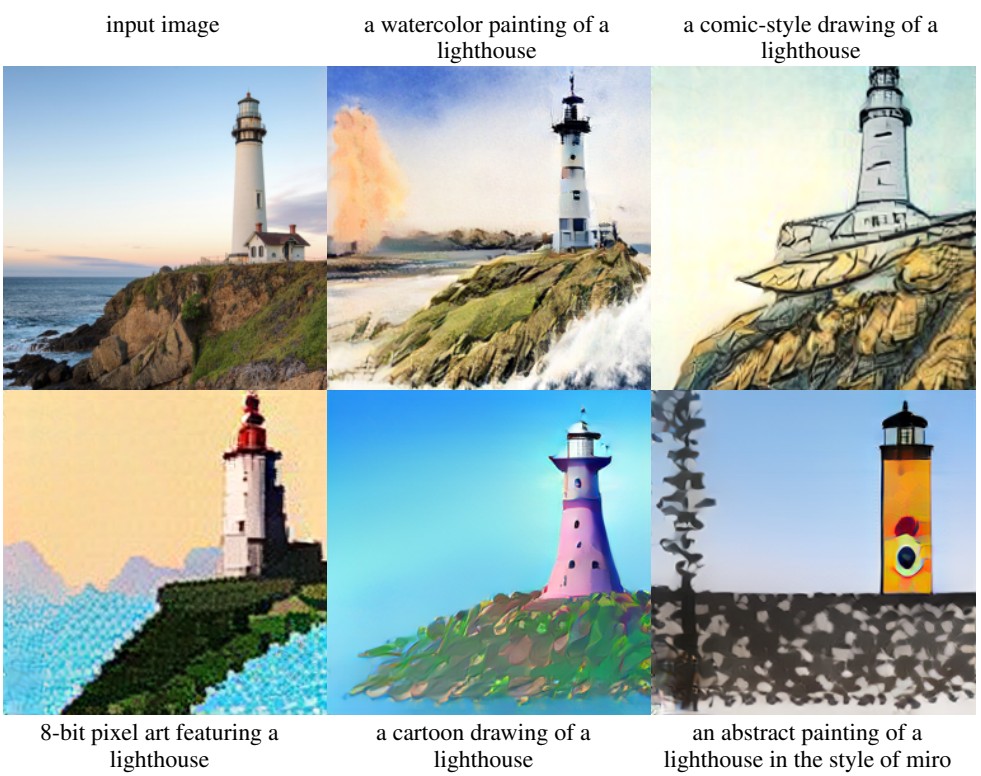

8-bit pixel art featuring a lighthouse   a cartoon drawing of a lighthouse   an abstract painting of a lighthouse in the style of miro

*Figure A8.* **Text-guided style editing to styles underrepresented in ImageNet.**

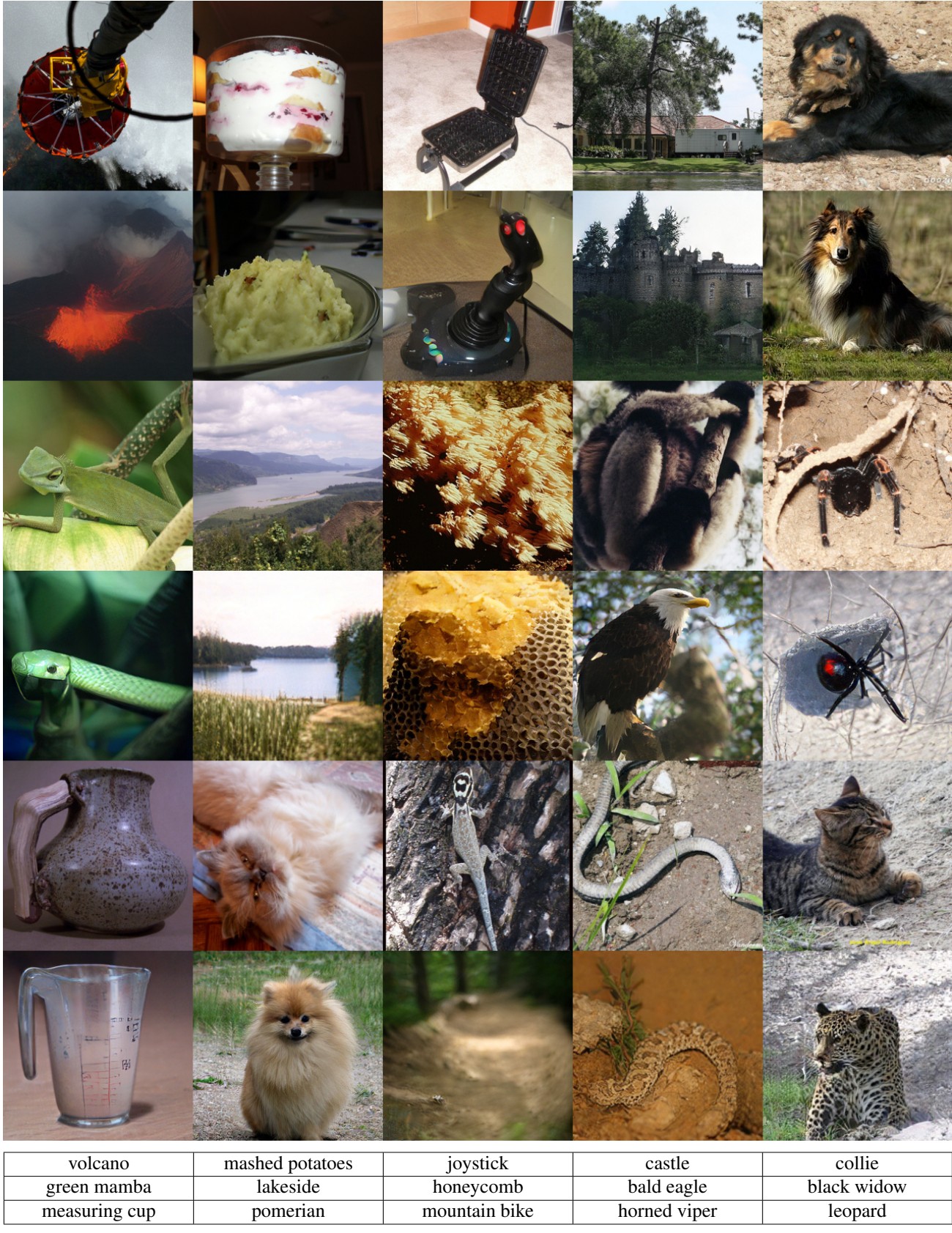

| volcano | mashed potatoes | joystick | castle | collie |
| --- | --- | --- | --- | --- |
| green mamba | lakeside | honeycomb | bald eagle | black widow |
| measuring cup | pomerian | mountain bike | horned viper | leopard |

*Figure A9.* **Selected generation results (rows 2, 4, 6)** alongside their seeds (rows 1, 3, 5). Corresponding classes shown in table.

