# OpenReview forum: "Highly Compressed Tokenizer Can Generate Without Training"
_ICML.cc/2025/Conference — ICML 2025 poster_

### Official Review · Reviewer_G4pF · 2025-02-24

**Overall Recommendation:** 3

**Summary:**

This paper proposes an optimization-based method to tweak the latent space of tokenizer for image editing tasks.

**Claims And Evidence:**

- I think this paper highlights the generation ability of the 1D tokenizer, specifically TiTok. However, I find using the word of "image editing" or "image variation" is better, because it is not like the standard generation model like diffusion model, that can generate from very cheap random noise, it needs to start from a seed image
- And on a high level, I feel this paper is telling two stories, i.e. section 3; section 4-5. It firstly shows the position 1D tokenizer has semantic meaning, and later presents the 1D tokenizer can be used to search the latent space. However, I don't see a clear connection between these two stories, or they are kind of broken. I think the reason of 1D tokenizer works better than 2D tokenizer for gradient-based method is its latent space is smaller and easier to perform optimization. It looks to me that even without the section 3, section 4-5 can already explain itself.
- I hope to see more failure cases and limitation discussed in the paper

**Essential References Not Discussed:**

Some other open-sourced 2D discrete image tokenizers:

- Autoregressive Model Beats Diffusion: Llama for Scalable Image Generation
- Cosmos World Foundation Model Platform for Physical AI

**Experimental Designs Or Analyses:**

- Regarding section 5, I wonder does 50k text prompts “a photo of a class” have duplicated prompts? Asking this because I think the optimization process is deterministic if you loss is deterministic and the number of iterations and optimizer are fixed. Then same text prompts should result in same image variation? Then I believe FID in this case won't be good? So I think if you want the FID low, you need to have non-duplicated promtps or make the optimization process have some randomness?

**Methods And Evaluation Criteria:**

.

**Other Comments Or Suggestions:**

.

**Other Strengths And Weaknesses:**

I think the design in section 3 is interesting, but I find it's a bit impractical to use because you always need to do a classification problem first to find the token that controls some concept.

**Questions For Authors:**

.

**Relation To Broader Scientific Literature:**

.

**Theoretical Claims:**

.

---

> ### Author Rebuttal · Authors · 2025-04-01
>
> Thank you for your review\! We hope to address some of your concerns in this response.
>
> * **Image Editing vs. Generation**
>
> We decided to choose the "image generation" nomenclature in line with previous work on so-called retrieval-augmented generative models, such as RDM \[1\].
>
> **Furthermore, as requested by Reviewer mQYF (see "Initializing with Random Tokens" for more details and quantitative results), we find that it is indeed possible to generate images "from scratch", by starting with random tokens.**
>
> Link to uncurated generations starting from random tokens: https://i.ibb.co/20L8xk1q/rand-tok.png
>
> * **Additional Failure Cases**
>
> The "from scratch" generations referenced above are uncurated, and we plan to include additional visualizations in the appendix of our revised submission.
>
> You may also be interested in our response to Reviewer apSr on additional out-of-domain examples.
>
> Finally, you can find a discussion of some limitations of the token optimization-based editing approach in our response to Reviewer N1ti's question regarding limited control over the editing process. In particular, there are no guarantees that text-guided editing will not result in unintended modifications to the input images, although we believe that this issue could be alleviated with further engineering effort (e.g., a more advanced objective function combining reconstruction and CLIP components).
>
> * **Determinism and Duplicated Seed/Prompt Inputs**
>
> All experiments in Section 5 do in fact include some small amount of randomness in the loss due to nondeterminism in some of the CUDA kernels used in our implementation \[2\].
>
> Interestingly, even the small amount of nondeterminism caused by implementation details of the CUDA kernels used in PyTorch is sufficient to lead to diverse generations by the test-time optimization procedure.
>
> We have experimentally verified that using deterministic implementations of these algorithms leads to deterministic results, in turn leading to degraded FID in the case of smaller number of seed images or deterministic seed selection (e.g. 500 seed images or 1000 seed images with top-1 selection). In the case of at least 1000 seed images with top-1% selection, there is enough diversity in the seed-to-prompt association to yield about 4000 unique inputs, such that the FID-5k results would be relatively unaffected by the use of a fully deterministic implementation.
>
> Thank you for pointing out this issue. We will address this point explicitly in the main text of the revised submission.
>
> ### References
>
> \[1\] Blattmann, A., Rombach, R., Oktay, K., Müller, J., and Ommer, B. Retrieval-augmented diffusion models. NeurIPS 2022\.
>
> \[2\] https://pytorch.org/docs/stable/generated/torch.use_deterministic_algorithms.html

---

### Official Review · Reviewer_N1ti · 2025-03-07

**Overall Recommendation:** 3

**Summary:**

The paper finds that tokens in the latent space of 1D tokenizers are strongly correlated with specific attributes of images (e.g. captured subject, lighting, background).
The authors build on this finding and propose gradient-based text-guided image editing and inpainting algorithms that optimize these 1D tokens.
This approach enables the use of pre-trained 1D image tokenizers for editing and inpainting without requiring further training.

**Claims And Evidence:**

The proposition of this paper is built on two main claims, as discussed below.

## Correlation between token positions and semantics

The authors claim that particular token positions in 1D tokenizers correlate with certain high-level attributes of images.
This is confirmed (for the pre-trained TiTok 1D tokenizer) through experiments in section 3, which analyze the encodings of various partitions of ImageNet. Specifically, the authors divide ImageNet into different classes (by utilizing CLIP similarity with a given prompt), and compute the variance (across classes) of the mean of tokens within classes (grouped by index).
Their results show that some token positions have a particularly high variance within a partition.

## Editing through manipulation of tokenizers

The authors show that meaningful image edits can be achieved through the manipulation of tokens at certain positions.
They experiment with perturbing tokens at certain positions (by replacing them with the tokens leading to the highest visual difference), which qualitatively lead to edits that "make sense" given the prompt with which the token index was chosen.
They also show that certain image attributes can be *transferred* from a reference to a target image by replacing a token of the encoded target image with that of the reference image at the desired token index.

**Essential References Not Discussed:**

The authors utilize test-time optimization on recent 1D image tokenizers for image editing and inpainting.
While the paper discusses other test-time optimization approaches for image editing, I recommend including a discussion on text-guided image editing approaches in a broader context, which would help clarify how the work relates to existing techniques.

**Experimental Designs Or Analyses:**

The authors present a series of experiments to compare seed sizes and seed association strategies.
They build on these results and claim that around 1000 seed images are enough to produce diverse generations. They also conclude that adding some stochasticity in the seed-to-prompt associations leads to better FID (as it increases diversity) but worse IS (compared to taking the associations having the best similarity).

- **eda.q.1** In line 370, "achieving an FID of 8.6 for 50k samples". Could the authors also provide the other metrics (IS, CLIP, SigLIP) for this experiment (and ideally add it to Table 1)?

The authors also present experiments that compare performance across different latent space dimensions, discrete and continuous tokens, as well as 1D and 2D tokenizers.

The authors conclude from Table 2 that a decreasing number of tokens leads to "significant improvements in generation quality". Upon further investigation, the authors compare VQ-LL-32, a **large size** variant of the tokenizer with 32 tokens, and VQ-BB-64, a **base size** variant of the tokenizer with 64 tokens (while keeping a constant codebook size).

- **eda.q.2** Could the authors provide additional evidence to confirm that the improvements indeed stem from the smaller number of tokens and not the larger model?

The authors conclude from Table 3 (first two rows) that a discrete latent space is essential in achieving good generative performance.
The authors also conclude from Table 3 (first and last row) that the large number of tokens in MaskGIT's VQGAN "prevent the successful application of the test-time optimization for generation" and that "the 2D tokenizers' spatially arranged tokens lead to optimization results that are spatially incoherent".

- **eda.q.3a** Is the degraded generative quality caused by the large number of tokens of the 2D tokenizer, the spatially arranged 2D grid, or both? It seems that further experiments are needed to support this claim, especially that eda.q.2 is not a given.

- **eda.q.3b** Regarding the spatially incoherent results, could the authors provide visuals that illustrate these observations?

The authors also provide some tweaks of their algorithm along with an ablation study, which justifies their design choices.

**Methods And Evaluation Criteria:**

## Methods

Building on the findings regarding the TiTok 1D tokenizer, the authors propose an image editing procedure that optimizes the encoded tokens of an image.
Specifically, they optimize encoded tokens to maximize a CLIP objective given a prompt.
However, key parts of the seed image (that are not related to the editing prompt) seem to change after the optimization (e.g. in Figure 5, the lighthouse itself seems to change between edits, although the prompt suggest changing its surrounding context).
This is expected as the suggested CLIP objective does not guarantee the conservation of elements unrelated to the editing.

Note that the authors claim in lines 271-274 that "the optimization preserves key aspects of the subject while aligning the generated image with the prompt". I would advise to reformulate this claim as key aspects of the subject **do change** between edits (e.g. the dog face in Figure 5, last row).

The authors also propose an inpainting procedure that utilizes a modified algorithm which periodically replaces the known parts of the image with their original counterparts, and encodes the resulting image back into the latent space.

- **mec.q.1.** How is the number of optimizer iterations chosen? In Figure 4, it seems that the higher the number of iterations, the further the resulting image deviates from the seed (and fewer elements of the seed image are conserved). Does running the optimization with a higher number of iterations result in the final image being completely different from the seed image? Could the authors provide the results of such an experiment?

- **mec.q.2.** In section 4.1, the authors say that optimizing $\mathbf{z}^{(k)}$ directly leads to poor results. Could the authors provide experiments that support this?

## Evaluation Criteria

The authors construct a "seed image" dataset that is subsampled from ImageNet.
They associate a target prompt to a seed image in various ways (random, top image in CLIP similarity, or randomly picked among top-k most similar images with CLIP).
They utilize various TiTok checkpoints, which allows them to compare results among different autoencoder types (with discrete or continuous tokens), different sizes, and different number of tokens.
They also compare with a 2D tokenizer (MaskGIT VQGAN).

To evaluate the diversity and quality of generation, the authors utilize the FID and IS metrics.
To evaluate alignment with target prompts, the authors utilize CLIP and SigLIP similarity.

**Other Comments Or Suggestions:**

## Writing suggestions

1. It would be helpful to provide equation numbers for future referencing.
2. In the equation of section 4.1, the text prompt is not represented and not given to the loss function (which might lead to confusion).
3. Typo in line 362 (second column).
4. Typo in line 375 (prevent**s**)

**Other Strengths And Weaknesses:**

## Strengths
1. The paper is well written and follows a clear and logical structure, making it an enjoyable and insightful read.
2. The concepts presented regarding 1D tokenizers in the paper are original, interesting, and insightful.
3. Additionally, the concepts presented hold potential for future research in this area.

## Weaknesses
1. My main concern is that some claims and conclusions that are important for the context of this paper require additional experimental support (see related questions).
2. Moreover, the test-time optimization text-guided editing algorithm seems to affect aspects of the image that are not related to the prompt (or that should be conserved). Could the authors elaborate on this issue?
3. Providing additional explanation and interpretation of the results, particularly in Section 5.3, would strengthen the discussion and findings of the paper (e.g. visualizations and insights on why a 2D latent space leads worse results).
4. For the sake of reproducibility, it would be valuable if the authors provided the code used to conduct the experiments in this paper as part of the supplementary material.

**Questions For Authors:**

Questions for the authors are listed in bullet points in their relevant context within the review.

**Relation To Broader Scientific Literature:**

This paper explores the latent space of one-dimensional image tokenizers which are relatively recent.
The authors highlight an interesting correlation between token positions and image semantics, and utilize it to perform editing and inpainting.
As such, this paper relates to other works involving 1D image tokenizers, and works tackling text-guided image editing and inpainting, specifically in a test-time optimization (training-free) context.

**Theoretical Claims:**

The authors do not make any theoretical claims worth discussing in this section.

---

> ### Author Rebuttal · Authors · 2025-04-01
>
> We thank you for your thorough and detailed comments and suggestions, as well as the thoughtful questions. We hope the following answers can address your concerns.
>
> * **mec.q.1.** Optimizer Iterations
>
> The assessment that the number of optimizer iterations will result in the input image deviating further and further from the input matches our results. In fact, the FID and IS scores are relatively sensitive to the number of iterations, and the choice of iterations can provide control over the FID/IS score tradeoff.
>
> We have therefore computed FID and IS for different number of optimizer iterations (between 50 and 500 in 50 iteration increments). We report FID and IS at the number of iterations corresponding to the best FID, and separately, to the best IS. *Note that these differ from the results in Table 2 because Table 2 reports results using the same number of optimizer iterations for all models.*
>
> |@ best FID|iter|FID|IS|
> |:-| :- | :- | :- |
> |VQ-LL-32|300|15.1|160|
> |VQ-BB-64|150|16.4|134|
> |VQ-BL-64|150|18.6|105|
> |VQ-BL-128|150|22.4|92|
>
> |@ best IS|iter|FID|IS|
> |:-|:-|:-|:-|
> |VQ-LL-32|500|15.5|175|
> |VQ-BB-64|250|17.1|144|
> |VQ-BL-64|250|19.1|118|
> |VQ-BL-128|250|24.5|95|
>
> For best results, the number of iterations should be picked on an example-by-example basis: "smaller" edits may require less iterations than more significant ones. One could also design adaptive stopping criteria based on monitoring the objective function's value and change across iterations.
>
> * **mec.q.2.** Optimizing without VQ
>
> We find that the vector quantization step serves as regularization that prevents the test-time optimization procedure from behaving akin to an adversarial attack on the CLIP objective. In particular, the CLIP objective, when provided with the output of the 1D tokenizer with VQ, appears to be very adversarially robust (see also [1]). We observe weaker robustness in the case without VQ.
>
> The best FID-5k achieved by the no-VQ optimization is 15.8 (at 100 optimization iterations, with an IS of 118) and the best IS is around 130 (with 250 optimization iterations, at an FID of 17.1). Even with L2 regularization applied on the tokens, we are not able to improve FID and see a very small improvement in IS. In all cases, the optimization with VQ significantly outperforms the no-VQ one with a top FID of 15.1 and IS of >160.
>
> We have generated examples of how this manifests itself qualitatively, which we will include in the revised submission. We observe that the no-VQ optimization sometimes leads to more artifacts and more exaggerated and larger areas of repetitive texture.
>
> **Link to visualization: https://i.ibb.co/3yYd9vF6/no-vq-vis.png**
>
> * **eda.q.1.** FID 8.6 Experiment
>
> We report this metric (FID-50k of 8.6) in Table 4, alongside the FID-5k and IS, CLIP and SigLIP scores. As you point out, IS, CLIP and SigLIP over 50k samples are not reported. We find that the metrics other than FID are very similar between the 5k and 50k sample evaluations, which is why we omit them.
>
> * **eda.q.2.** Decoder Size vs. Number of Tokens
>
> We agree that this is a shortcoming in our evaluation, so we have run additional experiments comparing the VQ-BL-64 and VQ-BL-128 tokenizers, which share the same model and codebook sizes, differing only in the number of 1D tokens. The results are included in the tables from our response to question **mec.q.1**, and show that the VQ-BL-64 tokenizer always outperforms the VQ-BL-128 tokenizer.
>
> * **eda.q.3a.** Role of 2D Tokens
>
> Our main reason for including the 2D VQGAN in this table was to demonstrate the poor performance of simple test-time optimization with widely used “traditional” tokenizers. Since we are not aware of any 2D tokenizers with VQ that achieve a similarly high compression ratio as TiTok, the conclusion from this experiment may be better reworded as *the high degree of compression enabled by 1D tokenization is key to generative performance*, rather than 1D tokenization itself being key.
>
> * **eda.q.3b.** 2D Tokens Qualitative Results
>
> Please see our response to Reviewer mQYF’s **Question 1** for some qualitative examples of 2D tokenizer generations.
>
> * **sm.q.1.** Code
>
> We will make the code publicly available.
>
> * **Preserving Input Image Features**
>
> There is indeed no guarantee on preservation of aspects of the input image.
>
> This could be mitigated with further engineering effort, e.g., the difference in tokens compared to the tokenized version of the input image could be slightly penalized. One could also combine the text-guided objective with the inpainting one to explicitly preserve user-defined regions.
>
> Certain differences wrt. the input are introduced by the tokenizer itself, due to imperfect reconstruction (VQ-LL-32 example: https://i.ibb.co/qMnp23fq/recons.png).
>
> ### References
>
> [1] S. Santurkar et al. Image synthesis with a single (robust) classifier. NeurIPS 2019. https://arxiv.org/abs/1906.09453

---

> > ### Comment · Reviewer_N1ti · 2025-04-03
> >
> > The authors have provided all the necessary clarifications and addressed my main concerns.
> >
> > Additionally, during the discussion period, the method appears to be **even more interesting** than initially thought, as it can generate (or "*edit*") images even starting from a random seed (and not just from an original seed image).
> >
> > I look forward to seeing the rebuttals' results reflected in the paper update.
> >
> > I still have some minor concerns about the non-preservation of key aspects of the input image and the number of iterations being variable and manually tuned depending on the example.
> >
> > Nevertheless, and in light of all of this, I have increased my Overall Recommendation. I would like to thank the authors for their diligent work during the rebuttal process !

---

### Official Review · Reviewer_apSr · 2025-03-12

**Overall Recommendation:** 2

**Summary:**

This paper introduces a generative pipeline leveraging a 1D image tokenizer (e.g., TiTok) to enable image editing and generation without training a dedicated generative model. By compressing images into highly compact 1D token sequences (e.g., 32 tokens), the authors demonstrate that simple token manipulations (e.g., copy-paste, gradient-based optimization) can achieve text-guided editing, inpainting, and unconditional generation. The approach relies on test-time optimization of tokens using objectives like CLIP similarity or reconstruction loss, bypassing the need for iterative generative models like diffusion. Experiments on ImageNet show competitive FID and IS scores compared to state-of-the-art methods, though qualitative results reveal limitations in handling complex scenes or novel concepts.

**Claims And Evidence:**

N/A

**Essential References Not Discussed:**

N/A

**Experimental Designs Or Analyses:**

N/A

**Methods And Evaluation Criteria:**

N/A

**Other Comments Or Suggestions:**

N/A

**Other Strengths And Weaknesses:**

**Strengths**

1. Novel Compression Approach: The use of 1D tokenization with extreme compression (32 tokens) is innovative, enabling efficient latent space editing and generation.

2. Training-Free Generation: The framework avoids training generative models, reducing computational overhead and enabling rapid adaptation to new tasks.

3. Empirical Validation: Comprehensive experiments (e.g., text-guided editing, inpainting) provide evidence of the method’s effectiveness, with competitive FID/IS scores on ImageNet.

4. Practical Applications: The approach’s flexibility could facilitate real-world use cases like content moderation, image editing, or low-resource generation.

**Weaknesses**

1. While the 1D tokenizer is novel, the core idea of latent space optimization (e.g., VQGAN-CLIP) is not groundbreaking. The contribution lies more in engineering than foundational advance.

2. Results are confined to ImageNet, leaving applicability to diverse domains (e.g., medical imaging, abstract art) unproven. Qualitative failures (e.g., Figure A6) highlight limitations with novel concepts or complex scenes.

3. FID/IS scores lag behind modern generative models (e.g., ADM, RCG-G), and qualitative outputs exhibit artifacts (e.g., blurriness in inpainting).

4. The paper lacks a rigorous analysis of why 1D tokenization enables generative capabilities, limiting its contribution to empirical observations.

5. Test-time optimization (300+ iterations) is computationally intensive, undermining claims of efficiency compared to trained generative models.

**Questions For Authors:**

N/A

**Relation To Broader Scientific Literature:**

N/A

**Theoretical Claims:**

N/A

---

> ### Author Rebuttal · Authors · 2025-04-01
>
> We thank you for your review! As you point out, the idea of latent space optimization is indeed not new. However, we do believe that its application in the case of highly compressed latent spaces is noteworthy for a few reasons:
>
> 1. **Previous attempts to use test time latent space optimization for image editing, i.e. VQGAN-CLIP, have not demonstrated high quality generation of photorealistic scenes, and rely on "tricks"**, such as the usage of a large number of different augmentations (various crops of the image being optimized, color-space corruptions, flips, etc.), in order to be successful. While these additional tricks can also be used to improve image generation in the case of highly compressed latent spaces, we demonstrate that they are not necessary. Remarkably, even an extremely straightforward application of test time optimization can produce reasonable results when operating in the highly compressed latent space (*baseline* in Table 4).
>
> 2. Since our baseline test time optimization algorithm is very simple (7 lines of code, see Algorithm A1 in the appendix), **we do not claim any significant engineering contribution**. Instead, we view our findings as strong motivation to view tokenizers with increasingly high compression ratios as generative models. In particular, we find it surprising that tokenizers trained with a standard VQGAN-like objective can be used to perform a variety of generative tasks such as text-guided image editing or inpainting. We would also like to point to our response to Reviewer mQYF, in which we provide experimental evidence that *the VQ-LL-32 tokenizer can even generate “from scratch”, starting from pure noise*.
>
> ---
>
> * **Comments on Out-of-Domain Generalization**
>
> Since TiTok tokenizers are trained only on ImageNet, we expect limited ability to generate complex out-of-distribution scenes, such as those involving classes that are not part of the ImageNet-1k dataset or requiring composition of multiple subjects (since ImageNet images often feature a single prominent subject). Unfortunately, no highly compressed 1D tokenizer trained on larger scale datasets was available at the time of submission, so our experiments are confined to ImageNet.
>
> However, as tokenizers with even higher compression ratios or trained on larger datasets become available, we expect application to more diverse domains to become possible. Further, we hope that the view presented in our paper – that **the lossy autoencoding task performed by tokenizers with very high compression ratios can be thought of as a generative modeling problem** – can provide an insightful perspective in scaling such tokenizers to these larger datasets.
>
> * **Additional Out-of-Domain Examples**
>
> As this was also requested by Reviewer mQYF, we generated additional qualitative results in the out-of-domain setting of text-guided style transfer. In particular, we use CLIP prompts like "a watercolor/pixel art/abstract/cartoon painting of a \<subject\>", and find that the model can produce qualitatively very compelling looking results for text-guided style transfer, even for styles which we expect to be mostly absent from the ImageNet-1k dataset. We will include the generations in the revised submission.
>
> **Link to visualization: https://i.ibb.co/Bd1HkmX/style.png**

---

### Official Review · Reviewer_dszF · 2025-03-14

**Overall Recommendation:** 3

**Summary:**

This work shows that a highly compressed 1D token set can learn different attributes in tokens, and perform generaion tasks such as inpainting and text-guided image editing with only a tokenizer, without any extra generative model training.

**Claims And Evidence:**

The claims are supported by experiments. The authors show disentangled attributes in tokens as well as the image editing and generation applications.

**Essential References Not Discussed:**

I did not find any missing key related works.

**Experimental Designs Or Analyses:**

In experiments, the authors show several interesting applications and visualizations for analysis.

**Methods And Evaluation Criteria:**

The method is simple and the findings are interesting. Evaluation is with standard metrics (FID / IS).

**Other Comments Or Suggestions:**

L290: top-1 (%) ?

**Other Strengths And Weaknesses:**

The attributes decomposition findings and training-free generation or editing are interesting.

However, the proposed method seems to be not directly scalable for the proposed applications. It relies on the properties of the compact tokens, which are more like emergent properties and not directly controllable.

**Questions For Authors:**

Is the gradient-based latent editing stable and can get high quality results with high probability?

**Relation To Broader Scientific Literature:**

This work is related to the recent work TiTok which converts images to a 1D set of tokens, and explores several interesting properties and applications with this idea of compact 1D tokenizer.

**Theoretical Claims:**

Theoretical proofs are not the focus in this work.

---

> ### Author Rebuttal · Authors · 2025-04-01
>
> Thank you for your review and helpful comments\!
>
> * **Per-Token Attributes as Emergent Properties**
>
> The direct token editing examples do indeed rely on emergent properties that are not controllable using the standard autoencoder-style/VQGAN training scheme used by TiTok.
>
> As such, we agree that practical applications of token copy-paste for image editing is restricted to a few tasks (such as brightness change, background blur, etc.), and these tasks have to be manually discovered using methods such as those described in Sections 3.1 and 3.2.
>
> However, we find the fact that such editing is possible at all very surprising. In particular, perturbation of individual tokens does not lead to semantically meaningful and globally coherent edits in the case of 2D tokenizers due to the spatial relationship of tokens and regions of the input image (see, for example, Figure 1 in \[1\]). We therefore intended to present this finding to draw attention to underexplored emergent properties of highly compressed latent spaces, as well as to motivate our more scalable gradient-based image editing approach from Section 4, which can be directly applied for practical applications such as text-guided editing or generation, or inpainting.
>
> * **Notation: Top-1/Top-1%**
>
> Regarding L290 (top-1 vs top-1%): Thank you for pointing out the confusing notation. "Top-1" is used to denote a seed image association procedure by which only the single most aligned seed image is chosen. In contrast, top-1% associates a random seed image from the subset of the top 1% most similar seed images (for example, with 1000 seed images, top-1% would correspond to random association to one of the top-10 images from the seed image pool). We will try to clarify this in the final version.
>
> * **Stability of Optimization Process**
>
> We do observe that the optimization process is sensitive to noise, such that even small perturbations to the objective and its gradients can lead to quite different generations. However, this does not mean that the generations are generally low quality – instead, we observe that this leads to generations that are diverse (even when adding only small amounts of noise), while still being reasonably high quality (as evidenced by our FID and IS scores).
>
> ### References
>
> \[1\] Cao, S., Yin, Y., Huang, L., Liu, Y., Zhao, X., Zhao, D., and Huang, K. Efficient-VQGAN: Towards high-resolution image generation with efficient vision transformers. ICCV 2023, https://arxiv.org/abs/2310.05400

---

### Official Review · Reviewer_mQYF · 2025-03-14

**Overall Recommendation:** 4

**Summary:**

The submission explores training-free image generation on TiTok's 1D tokenizer. It builds upon the observation that a heavily compressed tokenizer, like TiTok-L-32, is somewhat amenable to interpretable manipulation and editing of latents. The authors first demonstrate that by varying individual tokens, and by copy pasting latent manipulations from a reference example to a target image. They then build upon those insights and tune the tokens for an objective like CLIP score (between an image and prompt) with gradient descent through the TiTok decoder; i.e. without having to train a dedicated text-to-image model. Similarly, they are able to perform in-painting without training any model for that task.

## update after rebuttal
I thank the authors for their rebuttal. The conditional generation results from random initializations are quite interesting and I expect these results will strengthen the final paper. I will maintain my vote to accept this paper.

**Claims And Evidence:**

The claims in the paper are sufficiently supported by evidence. There are, however, some areas of uncertainty, which I list in the "Methods and Evaluation Criteria" section.

**Essential References Not Discussed:**

There are a few works that could be discussed in the larger context of "training-free" generation. One area is the "textual inversion" [1,2,3] line of research that optimizes one or multiple tokens to capture a concept, which can then be used to generate said concept. There is also a range of works that use CLIP as an optimization objective that could be discussed, e.g. see the ones listed in surveys like [4].

[1] An Image is Worth One Word: Personalizing Text-to-Image Generation using Textual Inversion, Gal et al., 2022

[2] Visual Lexicon: Rich Image Features in Language Space, Wang et al., 2024

[3] Training-Free Consistent Text-to-Image Generation, Tewel et al., 2024

[4] A Survey on CLIP-Guided Vision-Language Tasks, Yu et al., 2022

**Experimental Designs Or Analyses:**

The paper is not that detailed in terms of implementation details, but the methods are somewhat straight-forward and based on the descriptions I would feel comfortable reimplementing them to a good degree. Overall, the experimental design and analyses appear sound.

**Methods And Evaluation Criteria:**

I would say that most of the proposed evaluations make sense, and the visuals help motivate that the method can in fact generate images in a "training-free" manner. That said, there is a range of ablations that would be important to see:
1. How well does the proposed optimization procedure work when initializing with random tokens (instead of other images)? In other words, is this method limited to generating images from some existing seed image, or is it possible to perform the generation purely using the tokenizer's inductive biases and the optimization objective?
2. The paper shows results for 500-2k seed images, but how about choosing only a single (best) one? This goes into the direction of the question above, regarding the limitations of the optimization procedure.
3. Showing FID, IS, CLIP, and SigLIP scores is helpful, but I would be interested to see an analysis of how well a specific class can be generated, through the lens of a pre-trained classification model. CLIP and SigLIP scores show alignment, but the resulting scores are very close to each other after optimization, while there are much larger differences in FID and IS.
4. In Table 3, during optimization of the VAE model, did the authors apply the same KL-divergence, or KL-divergence between the optimized latents and real image latents? With discrete tokens, no token can individually be OOD, but in the VAE case it seems crucial to keep the soft constraints in mind. That then also goes into the analysis B.2.
5. How well does the proposed method compare to VQGAN-CLIP? This would be especially interesting as a comparison in Table 3.

**Other Comments Or Suggestions:**

The paper is easy to read and well motivated. I would suggest the authors to add more supporting visuals in the appendix.

**Other Strengths And Weaknesses:**

Overall, this is a very creative paper and a fun read! The paper is a callback to "early" CLIP-optimization-based image generation attempts. The fact that image generation can be performed in such a controlled and high-quality manner without any generative model training on top of the tokens is quite interesting, and the ablations show that such extreme compression ratios are necessary to achieve that (with the given optimization algorithm here). I also appreciate the attention of using SigLIP as an evaluation criteria and running the "adversarial" baseline.

**Questions For Authors:**

1. Visually speaking, how do the generations using the 2D tokenizer in Table 3 look?
2. For the ablation of continuous and 2D tokenizers, were they optimized using the vanilla objective, or did the authors try to regularize the latent space as well? (I.e. apply techniques such as presented in Table 4).
3. The additional qualitative results in Appendix D are interesting, and I would be glad to see more such OOD examples. Even though TiTok was trained on ImageNet-1k, it seems to generalize slightly for out-of-distribution images, which begs the question of how well this works on text-conditional generation in a larger domain.

**Relation To Broader Scientific Literature:**

The proposed method mostly builds upon the recent 1D tokenizer model TiTok, and first analyzes its latents and then proposes a training-free way to generate images using that tokenizer. For now it seems that the method is somewhat limited to such highly compressed latent spaces and does not work with 2D tokenizers, VAE variants, nor less compressed 1D tokenizers.

**Theoretical Claims:**

The paper does not make any theoretical claims.

---

> ### Author Rebuttal · Authors · 2025-04-01
>
> We are happy to hear you enjoyed our paper, and would like to thank you for the great questions, which we hope to answer below.
>
> * **Q1.** 2D Tokenizer Results
>
> We have produced visualizations of the optimization process using MaskGIT’s VQGAN, alongside the 32-token TiTok tokenizer, which we will include in the revised submission.
>
> **Examples from Figure A5: https://i.ibb.co/YB72k9Fh/2d-tok-vis-A5.png**
>
> **Additional example: https://i.ibb.co/WZnHmLx/2d-tok-vis.png**
>
> In this visualization, we observe that the highly compressed tokenizer allows the optimization procedure to perform globally consistent edits to the image. On the other hand, the VQGAN optimization is easily able to change local texture and color to align with the prompt, but fails to perform "global" edits (for example, when changing the species of an animal, the shape of the head and position of the ears are relatively unchanged in the VQGAN case, but attributes such as color can be adapted more successfully).
>
> **Regarding VQGAN-CLIP:** To the best of our knowledge, there is no existing evaluation of VQGAN-CLIP for ImageNet generation. We hope that our comparison of MaskGIT's VQGAN which is already included in Table 3 can provide such a comparison, as it should be very similar to VQGAN-CLIP while being fair in the sense that the same seed image selection and cost function are used as for the 1D tokenizer experiments.
>
> * **Q2.** Token Regularization
>
> **VAE Tokens.** The experiment in the paper did not include regularization on the VAE tokens. We agree this is crucial, and have repeated this experiment. Results are provided in the table below.
>
> |  |FID-5k|IS|CLIP|SigLIP|
> |:-|:-|:-|:-|:-|
> |**without L2 reg.**|39.6|66|0.48|2.66|
> |**with L2 reg. (weight=0.2)\*** |33.2|93|0.46|3.05|
>
> \*weight chosen for best results from sweep including {0.02, 0.1, 0.2, 0.5}
>
> While the results for VAE tokens with regularization are improved over the numbers reported in the original submission, FID and IS are still poor compared to the VQ model, so we do not change our conclusion regarding the importance of VQ.
>
> **2D Tokens.** For MaskGIT’s VQGAN, we do not use regularization in the experiment from Table 3. We do not find that it makes a substantial difference.
>
> * **Q3.** Additional OOD Examples
>
> We have run an additional out-of-domain text guided style-transfer task, which we describe in our response to Reviewer apSr.
>
> * **Initializing with Random Tokens**
>
> We decided to run some additional experiments and find that it is indeed possible to generate images "from scratch", starting from randomly sampled tokens!
>
> While results are worse than in the case of CLIP-based prompt-to-seed association, FID and IS scores are still reasonable. Qualitatively, we find that it is still possible to generate relatively high quality samples, especially with more detailed CLIP prompts and longer optimization times (400-500 iterations). We plan to include uncurated generations starting from randomly sampled tokens (with **no** seed image) in the revised submission.
>
> **Link to uncurated generations starting from random tokens: https://i.ibb.co/20L8xk1q/rand-tok.png**
>
> Quantitative results (using the same settings as the line marked with (\*) from Table 4 in the paper) can be found in the table below:
>
> |  |iters|Seed Assoc|FID-5k|IS|CLIP|SigLIP|
> |:-|:-|:-|:-|:-|:-|:-|
> | **start from seed image** | 300 | CLIP-top-1% | 15.1 | 160 | 0.39 | 2.53 |
> | **start from random tokens\*** | 300 | n/a | 17.0 | 114 | 0.38 | 1.68 |
>
> **With** additional tweaks from Table 4 (token L2 regularization, extra token noise in the case of starting from a seed image, extra 100 iterations in the case of starting from random tokens), the gap in performance between the seed-image-initialized and random-token-initialized optimizations diminishes further:
>
> |  | iters | Seed Assoc | FID-5k | IS | CLIP | SigLIP |
> | :---- | :---- | :---- | :---- | :---- | :---- | :---- |
> | **start from seed image** | 300 | CLIP-top-1% | 14.6 | 182 | 0.38 | 2.07 |
> | **start from random tokens\*** | 400 | n/a | 15.5 | 152 | 0.40 | 2.54 |
>
> *\*For both tables, tokens are randomly initialized from normal distribution with std=0.3, chosen as the best performing value from a sweep including {0.05, 0.3, 1.0}.*
>
> **Note:** Starting from random tokens leads to **better performance** than starting from random seed images (i.e., without CLIP-based seed-to-prompt association) – cf. Table 1 row 3 (line 284).
>
> * **Initializing from Single Best Image**
>
> In Table 1 (L290), we show the results of an experiment where only the best seed image from the seed image pool is picked for each prompt. This allows achieving high IS at the expense of degraded FID.
>
> If you are referring to picking only one overall best image, and using that same image for every prompt, we suspect performance may not be great, since our previous experiment initializing with random tokens can achieve better results than initializing with random images.

---

### Decision · Program_Chairs · 2025-05-01

**Decision:**

Accept (poster)

**Comment:**

- Reasons to Accept:
Novel application of 1D tokenizers for generation and editing (agreed by all reviewers).
Training-free approach is simple, elegant, and surprisingly effective (noted by mQYF, N1ti).
Strong empirical results with competitive FID/IS scores and compelling ablations (mQYF, apSr).
Rebuttal provided new experiments on generation from random tokens and clarified key limitations (N1ti upgraded his score).

- Reasons to Reject / Concerns:
Limited generalization beyond ImageNet and reliance on 1D tokenizer properties (apSr, G4pF).
Editing instability and lack of control over preserved image regions (dszF, N1ti).
Disconnection between token analysis (Sec. 3) and generation pipeline (Sec. 4–5) (G4pF).
Contribution is mostly empirical (apSr).

Overall even if simple the observation has been considered as interesting by most reviewer and is worth to be discussed.